# Loss of Dnmt3a and Dnmt3b does not affect epidermal homeostasis but promotes squamous transformation through PPAR-γ

Lorenzo Rinaldi[1,2,3], Alexandra Avgustinova[1], Mercè Martín[1], Debayan Datta[1], Guiomar Solanas[1], Neus Prats[1], Salvador Aznar Benitah[1,4]*

[1]Institute for Research in Biomedicine (IRB Barcelona), The Barcelona Institute of Science and Technology, Barcelona, Spain; [2]Centre for Genomic Regulation, The Barcelona Institute of Science and Technology, Barcelona, Spain; [3]Universitat Pompeu Fabra, Barcelona, Spain; [4]Catalan Institution for Research and Advanced Studies (ICREA), Barcelona, Spain

**Abstract** The DNA methyltransferase Dnmt3a suppresses tumorigenesis in models of leukemia and lung cancer. Conversely, deregulation of Dnmt3b is thought to generally promote tumorigenesis. However, the role of Dnmt3a and Dnmt3b in many types of cancer remains undefined. Here, we show that Dnmt3a and Dnmt3b are dispensable for homeostasis of the murine epidermis. However, loss of Dnmt3a-but not Dnmt3b-increases the number of carcinogen-induced squamous tumors, without affecting tumor progression. Only upon combined deletion of Dnmt3a and Dnmt3b, squamous carcinomas become more aggressive and metastatic. Mechanistically, Dnmt3a promotes the expression of epidermal differentiation genes by interacting with their enhancers and inhibits the expression of lipid metabolism genes, including PPAR-γ, by directly methylating their promoters. Importantly, inhibition of PPAR-γ partially prevents the increase in tumorigenesis upon deletion of Dnmt3a. Altogether, we demonstrate that Dnmt3a and Dnmt3b protect the epidermis from tumorigenesis and that squamous carcinomas are sensitive to inhibition of PPAR-γ.

*For correspondence: salvador.aznar-benitah@irbbarcelona.org

**Competing interests:** The authors declare that no competing interests exist.

## Introduction

DNA methylation is an epigenetic mechanism that regulates several aspects of gene expression, such as long-term gene silencing, transcriptional elongation, and maintenance of genomic stability (*Allis and Jenuwein, 2016*; *Avgustinova and Benitah, 2016*; *Rinaldi and Benitah, 2015*). It is found throughout the vertebrate genome and is deposited by DNA methyltransferases on the fifth position of cytosine (5-mC), predominantly at CpG dinucleotides. The role of DNA methylation in establishing different cell fates during embryogenesis is fairly well understood. However, if and how DNA methylation is necessary to stably maintain the identity of adult stem cells, and how this process is disrupted during oncogenic transformation, is under intense investigation (*Shen and Laird, 2013*).

Three DNA methyltransferases are encoded in the vertebrate genome. Dnmt1 is predominantly associated with the maintenance of DNA methylation following cell division due to its high affinity for hemimethylated DNA. Consequently, depletion of Dnmt1 leads to a significant reduction of the global levels of 5-mC (*Li et al., 1992*; *Lei et al., 1996*). Dnmt3a and Dnmt3b are de novo DNA methyltransferases that establish genome-wide DNA methylation during mammalian embryogenesis and adult stem cell homeostasis (*Okano et al., 1999*). In mouse embryonic stem cells, the combined

**eLife digest** Most of the cells in our body contain the same DNA. However, our bodies are made of many different types of cell, such as nerve cells or skin cells, which perform very different jobs. In each cell type only certain sets of genes encoded by the DNA are active. Proteins known as epigenetic regulators are responsible for producing the different patterns of gene activity. If epigenetic regulators are switched on or off at the wrong time, they can contribute to ageing and diseases such as cancer.

Enzymes known as DNA methyltransferases are one group of epigenetic regulators. DNA methyltransferases control the activity of genes by adding small chemical groups known as methyl groups to the DNA. Two of these enzymes – known as Dnmt3a and Dnmt3b – are important during development to help cells mature and specialize into different types. Mice that lack both of these enzymes either die as embryos or just after birth. Furthermore, these enzymes are mutated or less active in some skin cancers and various other human cancers.

Here, Rinaldi et al. investigated the role these enzymes play in adult mice. The experiments show that under ordinary laboratory conditions, mutant mice that lacked Dnmt3a and Dnmt3b were as healthy as normal mice. However, when the mice were exposed to chemicals that promote tumor growth, which mimics skin exposure to UV light, the mutant mice developed many more skin tumors than the normal mice. Furthermore, the tumors in the mutant mice were more likely to form secondary tumors in the lung. Rinaldi et al. found that Dnmt3a reduced the production of a protein called PPAR-γ, which helps to break down some types of fat molecules. Treating the mutant mice with a drug that inhibits PPAR-γ activity slowed the growth of the tumors.

Overall, these experiments show a new way in which DNA methyltransferases act in adult animals. Future research will investigate whether drugs that inhibit the breakdown of fats could help to treat cancers in which the Dnmt3a and Dnmt3b proteins are mutated or less active.

loss of Dnmt3a and Dnmt3b leads to the progressive loss of DNA methylation, suggesting that these enzymes are additionally involved in maintaining 5-mC levels (*Chen et al., 2003*).

Since Dnmt3a-null mice die perinatally, and ablation of Dnmt1 and Dnmt3b results in embryonic lethality, conditional deletion mouse models have been necessary to study their functions in adulthood (*Ueda et al., 2006*; *Okano et al., 1999*). Hematopoietic stem cells (HSCs) lacking Dnmt3a cannot differentiate correctly upon serial transplantation, and end up developing a range of severe myeloid and lymphoid malignancies in aged animals (*Challen et al., 2011*; *Mayle et al., 2015*). Conversely, HSCs-depleted of Dnmt3b show no phenotypical differences with respect to wild-type controls, whereas combined ablation of Dnmt3a and Dnmt3b in HSCs result in an enhanced block of hematopoietic differentiation as compared to Dnmt3a loss alone (*Challen et al., 2014*). Interestingly, the observed phenotype seems specific to stem cells, as the fully differentiated cardiac myocytes carrying a combined deletion of Dnmt3a and Dnmt3b are indistinguishable from wild-type controls (*Nührenberg et al., 2015*). Similarly, to HSCs, purified murine neural stem cells (SCs) lacking Dnmt3a do not show problems with self-renewal but fail to differentiate properly (*Wu et al., 2010*). In addition, Dnmt3a acts as a potent tumor suppressor in lung tumorigenesis, to promote adenoma progression but not initiation, downstream of oncogenic K-Ras (*Gao et al., 2011*). This is in contrast with the pro-tumorigenic activity of Dnmt3b, which at least in the murine intestinal epithelium cooperates with the loss of APC to drive adenoma initiation and growth (*Steine et al., 2011*; *Lin et al., 2006*)

Recently, progress has been made in identifying the molecular mechanisms underlying the biological functions of Dnmt3a and Dnmt3b by studying their genome-wide localization. For instance, Dnmt3b associates with and methylates the gene bodies of actively transcribed genes in murine embryonic SCs and human embryonic carcinoma cells (*Baubec et al., 2015*; *Jin et al., 2012*; *Morselli et al., 2015*). Likewise, it has been proposed that gene body methylation is responsible of most of the transcriptional changes underlying the ability of Dnmt3a to promote neural SCs differentiation, and in protecting the lung epithelium from tumor progression (*Wu et al., 2010*; *Gao et al., 2011*).

We have recently reported that Dnmt3a and Dnmt3b are required for the self-renewal of human keratinocyte progenitors, whereas Dnmt3a is also required for their proper differentiation (*Rinaldi et al., 2016*). Mechanistically, Dnmt3a and Dnmt3b bind to and promote the activity of enhancers in both human epidermal progenitors and differentiated keratinocytes (although Dnmt3a having a stronger affinity for enhancers in differentiated keratinocytes). Interestingly, both proteins preferentially associate to super-enhancers rather than typical enhancers. Nonetheless, they differ in their mechanism of action, since Dnmt3a (together with Tet2) is essential to maintain high levels of 5-hydroxymethylcytosine (5-hmC) at the center of its target enhancers, while Dnmt3b promotes 5-mC along the body of the enhancer. These regulatory regions dictate the transcription of essential genes necessary for epidermal stem cell identity and maintenance, such as *FOS, ITGA6, TP63,* and *KRT5*. Similar to its role in mouse ES cells, Dnmt3b also binds to and methylates the gene bodies of these genes to reinforce their expression (*Rinaldi et al., 2016*). Dnmt3a also associates to the enhancers regulating the expression of genes such as IVL, LOR, FLG2, and KRT1 which drive the differentiation of SCs into mature keratinocytes (*Rinaldi et al., 2016*). Interestingly, DNA methylation at active enhancers has also been recently reported in normal and cancer-derived human colon, mammary and prostate epithelial cells (*Charlet et al., 2016*).

However, to date, no in vivo studies have investigated the roles of Dnmt3a and Dnmt3b in adult epidermal function and malignant transformation. Using mouse models carrying an epidermis-specific ablation of either Dnmt3a or Dnmt3b, or both, we demonstrate that Dnmt3a and Dnmt3b are dispensable for skin homeostasis. However, Dnmt3a has a critical role in suppressing carcinogen-induced squamous tumor initiation, but not progression, while both Dnmt3a and Dnmt3b concertedly prevent tumor progression.

## Results

### Epidermal deletion of Dnmt3a does not affect steady-state tissue homeostasis but significantly promotes carcinogen-induced tumorigenesis

We first studied the pattern of expression of Dnmt3a and Dnmt3b during epidermal development, and in adulthood. At E14.5, Dnmt3a was expressed in the entire Keratin-14+ compartment comprising the basal layer of the embryonic epidermis and the hair placodes (*Figure 1—figure supplement 1A*). At P0, all Keratin-14+ basal cells were positive for Dnmt3a with the exception of the highly proliferative hair follicle bulb cells (*Figure 1—figure supplement 1A–B*). By the time animals reached adulthood, Dnmt3a levels remained high in the hair follicle bulge where most hair follicle stem cells reside (*Solanas and Benitah, 2013*), and decreased in the interfollicular epidermis, although some basal IFE cells expressed high levels (*Figure 1—figure supplement 1A–D*). On the other hand, we were not capable of detecting Dnmt3b by immunofluorescence staining in sections of developing or adult mouse epidermis (not shown), suggesting that Dnmt3b is expressed at low levels (*Challen et al., 2014*). In fact, RNA-seq data confirmed that Dnmt3a was enriched almost fivefold as compared to Dnmt3b in adult basal IFE keratinocytes (*Figure 1—figure supplement 1C*). However, Dnmt1, the main DNA methyltransferase, was the most abundant DNA methyltransferase, both in interfollicular epidermis and in hair follicle bulge cells (*Figure 1—figure supplement 1C*).

To gain insight to the roles of Dnmt3a and Dnmt3b in epidermal tissue function, we generated epidermis-specific conditional knockout (cKO) mice by crossing animals containing the *Dnmt3a* or *Dnmt3b* gene flanked by loxP sites with animals carrying the *Keratin14-CRE-ROSA26*-YFP-cassette (hereafter referred to as Dnmt3a/3b-cKO) (*Gao et al., 2011*). Surprisingly, neither Dnmt3a- nor Dnmt3b-cKO displayed noteworthy epidermal phenotypical differences as compared to their wild-type littermates at different postnatal ages (*Figure 1—figure supplement 1D–E* and *Figure 2—figure supplement 1A*). Likewise, despite its strong expression in hair follicle stem cells, the loss of Dnmt3a did not result in evident changes in hair follicle cycling and pelage growth (*Figure 1—figure supplement 1E–F*).

Deregulation of DNA methylation can alter gene expression, leading to tumor suppressor silencing or oncogene activation (*Witte et al., 2014*), and mutation/deregulation of Dnmt3a and Dnmt3b has been observed in several tumor types (*Leppert and Matarazzo, 2014*; *Subramaniam et al., 2014*). Recently, Dnmt3a has attracted much attention, as it is one of the most frequently mutated

genes in cancer (*Kim et al., 2013*), especially in acute myeloid leukemia (*Garg et al., 2015*; *Ley et al., 2010*). In fact, a loss-of-function mutation of Dnmt3a is one of the earliest mutations that occurs in human acute myeloid leukemia (*Shlush et al., 2014*). Importantly, these mutations are functional since knock-in mice that model it develop a range of severe myeloid and lymphoid malignancies (*Challen et al., 2011*; *Mayle et al., 2015*). In addition, HSCs harboring inactivating mutations of Dnmt3a are clonally selected in ageing humans (*Shlush et al., 2014*). However, much less is known about how deregulation of Dnmt3a and Dnmt3b affect tumorigenesis in epithelial tissues.

To elucidate the roles of Dnmt3a and Dnm3b in skin tumorigenesis, we first generated tumors from the epidermis using the chemically induced carcinogenesis protocol based on DMBA/TPA (*Ewing et al., 1988*). The first epidermal squamous malignancies appeared significantly sooner in Dnmt3a-cKO than in their wild-type littermates, after only 2 months from the first DMBA treatment, indicating that Dnmt3a acts as a barrier against tumor initiation (*Figure 1A,B*). Dnmt3a-cKO animals also showed a significant increase in tumor burden, with an average of 17 tumors per animal compared to three tumors per wild-type animal after 6 months of initiating the experiment (*Figure 1C* and *Figure 1—source data 1*).

Although Dnmt3a-cKO animals showed a strong increase in tumor initiation and burden, they developed the same percentage of squamous cell carcinomas than wild-type mice (*Figure 1D,E*). Indeed, a detailed histological analysis of the tumors collected from Dnmt3a-cKO and wild-type animals indicated that Dnmt3a-cKO mice developed the same percentage of benign tumors, such as keratoacanthomas and papillomas, as well as of malignant invasive papillomas and squamous cell carcinomas (SCCs) (*Figure 1E*). Dnmt3a-cKO mice only developed an increase in the percentage of sebaceous adenomas (*Figure 1E*). No metastases were scored in any of the animals, as expected using this protocol in mice with a C57/Bl6 genetic background (*Sundberg et al., 1997*). Altogether, these results indicate that loss of Dnmt3a dramatically increases initiation of epidermal squamous tumors without affecting their malignant progression, and slightly skews the histology of tumors towards the sebaceous lineage.

## Deletion of Dnmt3b does not increase tumor initiation, but synergizes with loss of Dnmt3a to promote tumor progression and metastasis

Dnmt3a suppresses K-Ras-driven lung tumor progression, whereas Dnmt3b is pro-tumorigenic in APC-deficient colorectal adenomas (*Gao et al., 2011*; *Lin et al., 2006*). Hence, we next tested whether Dnmt3a and Dnmt3b also exert opposing effects regarding tumorigenesis in the epidermis. Interestingly, there were no differences between wild-type and Dnmt3b-cKO mice with respect to either the timing of tumor initiation or tumor burden upon treatment with DMBA/TPA (*Figure 2A*-right panel). There were no significant changes in the histological appearance of the tumors, or the number of basal cells proliferating or undergoing apoptosis, between Dnmt3b-cKO and the wild-type controls (*Figure 2—figure supplement 2* and *Figure 2—figure supplement 3A–B*).

To assess whether Dnmt3a and Dnmt3b potentially play redundant roles during tumorigenesis, we also induced tumors using the DMBA/TPA protocol in animals carrying an epidermis-specific deletion for both Dnmt3a and Dnmt3b in combination (DcKO). Strikingly, although DcKO animals had a severe depletion of DNA methylation in their epidermises, they formed a morphologically normal skin with all its appendages and did not develop any epidermal abnormality even up to 70 weeks of age (*Figure 2—figure supplement 4A–B*). This strongly suggests that de novo DNA methylation is dispensable for the long-term homeostasis of undamaged epidermis. However, when subjected to tumorigenesis, DcKO animals displayed a significantly shortened latency and significant higher tumor burden than wild-type mice (*Figure 2B–F*). Although these differences were similar to the ones observed in single Dnmt3a-cKO mice (*Figure 2C–D*), DcKO mice formed aggressive squamous cell carcinomas at a higher frequency as compared to the single cKOs of Dnmt3a or Dnmt3b (*Figure 2E*). In addition, metastatic nodules in the lungs were observed in 30% of DcKO animals, but in none of the wild-type, Dnmt3a-cKO, or Dnmt3b-cKO animals (*Figure 2F*). Recent reports show that epidermal squamous cell carcinomas that harbor cells undergoing epithelial to mesenchymal transitions are more metastatic than those that remain predominantly epithelial in nature (*Latil et al., 2017*; *da Silva-Diz et al. 2016*). Interestingly, DcKO tumors contained large areas with spindle-shaped cells that expressed lower levels of the epithelial markers E-Cadherin and Keratin14, compared to the wild type and to Dnmt3a-cKO tumors (*Figure 2E*, and *Figure 2—figure supplement 5*). These cells also expressed the mesenchymal marker Vimentin (*Figure 2—figure*

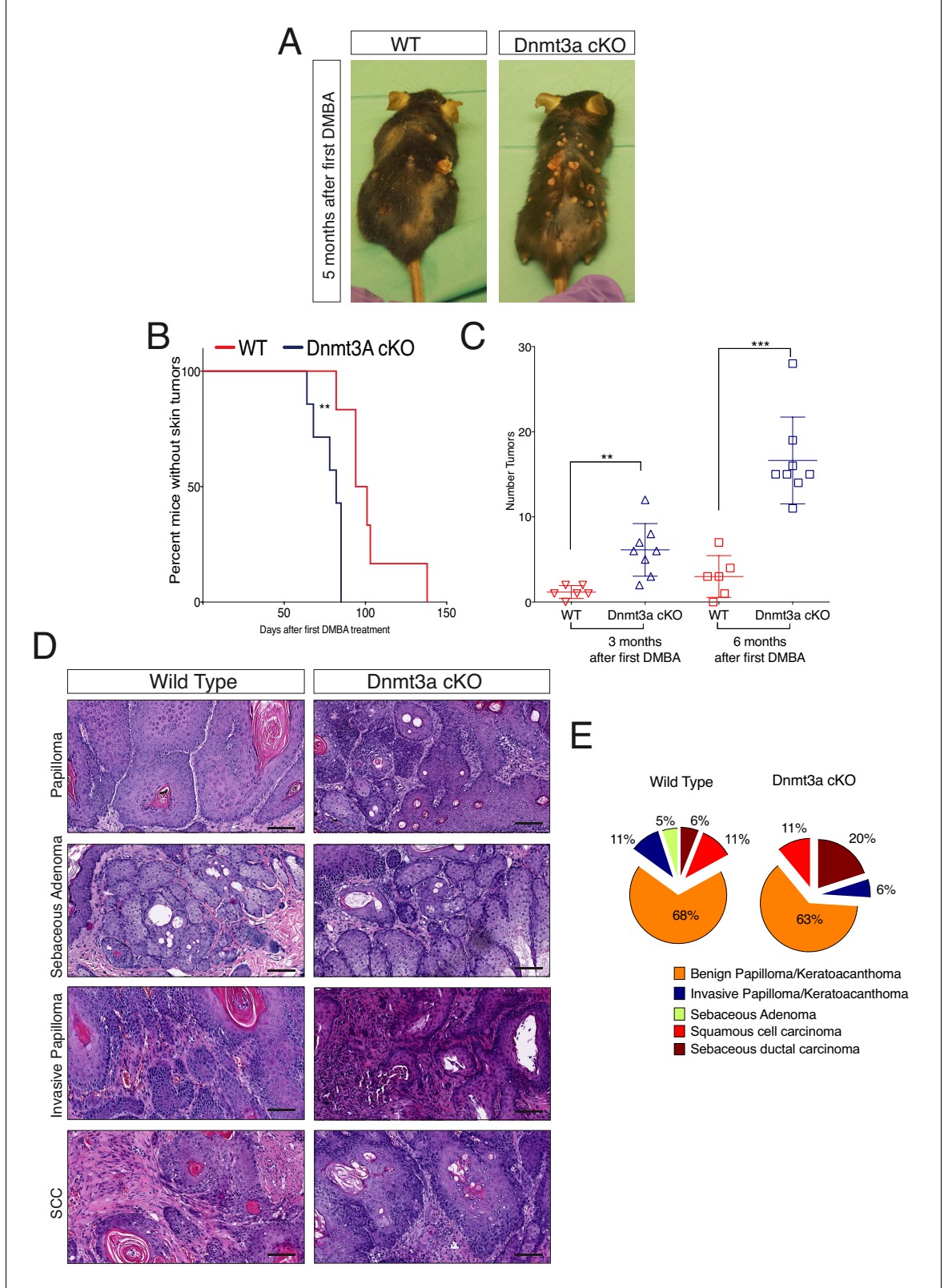

**Figure 1.** Dnmt3a loss shortens the onset of carcinogen-induced skin neoplasia, and increases tumor burden. (**A**) Representative pictures of wild-type and Dnmt3a-cKO animals after 5 months of treatment with DMBA/TPA. Graph in panel A represents the percentage of animals WT (n = 6) or Dnmt3a-cKO (n = 6) that entered into anagen after 2 weeks of treatment of DMBA/TPA, p=0.02, Chi-Square test. (**B**) Time of appearance, expressed in percentages of skin tumors on wild-type or Dnmt3a-cKO animals, p=0.005. (**C**) Number of skin tumors after 3 or 6 months of DMBA/TPA treatment,

*Figure 1 continued on next page*

Figure 1 continued

p=0.001 and p=0.0007. (D) Representative images (hematoxylin/eosin staining) of different subtypes of skin tumors. (E) Histopathological analysis of the different subsets of skin tumors that appeared after DMBA/TPA treatment of wild-type or Dnmt3a-cKO animals.

The following source data and figure supplement are available for figure 1:

**Source data 1.** Data related to *Figure 1B* and *Figure 1C*.

**Figure supplement 1.** Dnmt3a is highly expressed in the basal cells of the interfollicular epidermis (IFE), and in the bulge of hair follicles in young mice.

*supplement 6A–C*). Importantly, these cells that had undergone a mesenchymal transition were still YFP+, thus deriving from the K14+ origin of the tumor (*Figure 2—figure supplement 6*).

Taken together, these results indicate that Dnmt3a and Dnmt3b are dispensable for epidermal homeostasis, and that Dnmt3a, but not Dnmt3b, suppresses skin squamous tumor initiation. However, both Dnmt3a and Dnmt3b repress the malignant transformation of epidermal cells into aggressive squamous cell carcinomas.

## Deletion of Dnmt3a results in increased tumor transcriptome heterogeneity and upregulation of genes related to lipid metabolism

We next wanted to characterize the molecular mechanisms that might underlie the tumor-suppressive function of Dnmt3a in the epidermis. To this end, we isolated by FACS-based cell sorting the basal integrin α6$^{bright}$ tumor cells from four wild-type and eight Dnmt3a-cKO tumors, and performed whole-genome transcriptome profiling by RNA-seq (*Figure 3A*). It is important to note that our mouse pathologists scored these tumors as squamous cell carcinomas (SCCs), and that ADFP (Perilipin-2) expressing sebaceous adenomas were not included in this transcriptome study (*Figure 3—figure supplement 1A–C*).

The PCA analysis of the RNA-seq samples showed that the four wild-type tumors clustered together, indicating that overall their transcriptomes were defined by common genes (*Figure 3B,C*). In contrast, the transcriptomes of Dnmt3a-cKO tumors were substantially more heterogeneous, suggesting that the loss of Dnmt3a could result in the deregulation of numerous different pathways in cancer cells, or that in the context of Dnmt3a loss, different cell of origins (i.e. basal IFE cells, hair follicle stem cells, or Lrig+ stem cells) might be more prone to generate more transcriptionally divergent tumors. Nevertheless, 391 genes were consistently differentially expressed between wild-type and Dnmt3a-cKO tumors, of which 114 were downregulated and 277 were upregulated in the latter (*Supplementary file 1*). The downregulated genes were mainly associated with apoptosis, suggesting that loss of Dnmt3a promotes cell survival and protects against programmed cell death (*Figure 3D*); consequently, TUNEL and active Caspase3 staining confirmed that Dnmt3a-cKO tumors had fewer apoptotic cells as compared to wild-type tumors (*Figure 3—figure supplement 2A–B* and *Figure 3—figure supplement 2—Source data 1*). Dnmt3a-cKO tumors also expressed higher levels of several genes involved in cell proliferation (*Figure 3D*). Interestingly, proliferation was only significantly increased in pre-cancerous DMBA/TPA-treated Dnmt3a-cKO epidermis (*Figure 3—figure supplement 3A–B*), while no differences in proliferation were evident between the homeostatic epidermis and tumors of wild-type and Dnmt3a-cKO mice (*Figure 3—figure supplement 3A–B*). Altogether, this suggests that the loss of Dnmt3a endows pre-cancerous mutant basal cells with a survival and proliferative advantage, which could account for the increased number of tumors these mice develop. However, once tumors are formed, they progress with the same kinetics as wild-type tumors (*Figure 3—figure supplement 3C* and *Figure 3—figure supplement 4—Source data 1*).

Gene ontology (GO) analysis of the 277 genes that were upregulated in Dnmt3a-cKO basal tumor cells highlighted two principal pathways that were over-represented in all eight Dnmt3a-cKO tumors: Wnt signaling (including ligands and receptors), and more predominantly, lipid metabolism (*Figure 3D*). Interestingly, recent reports have associated an increase in lipid metabolism with increased tumorigenesis of chronic myeloid leukemia, as well as colorectal, liver, oral, and breast cancer (*Beyaz et al., 2016*; *Ma et al., 2016*; *Camarda et al., 2016*; *Ye et al., 2016*; *Corbet et al., 2016*; *Schug et al., 2015*; *Pascual et al., 2017*; *Wahl et al., 2017*; *Bensaad et al., 2014*; *Luo and Puigserver, 2016*). A number of genes associated with fatty acid and lipid metabolism were

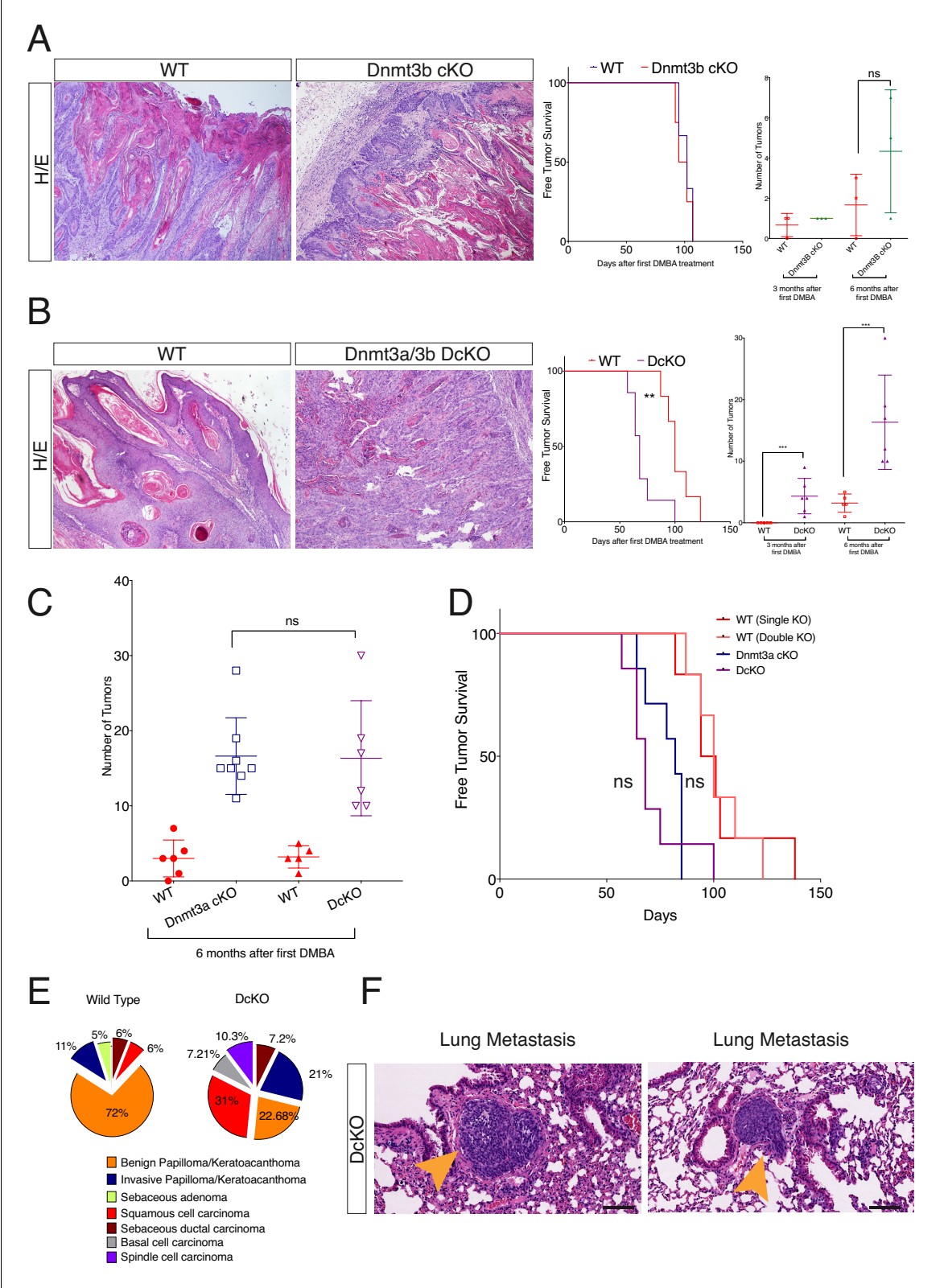

**Figure 2.** Dnmt3a and Dnmt3b double cKO animals develop more aggressive tumors than wild-type, Dnmt3a-cKO and Dnmt3b-cKO mice. (**A**) Left, representative images (hematoxylin/eosin staining) of skin tumors isolated from wild type and Dnmt3b-cKO littermates after 6 months of DMBA/TPA treatment. Right, time of appearance of tumors shown as percentages in wild-type and Dnmt3b-cKO animals, and number of skin tumors after 3 or 6 months of DMBA/TPA treatment. (**B**) Left, representative images (hematoxylin/eosin staining) of skin tumors isolated from wild type and Dnmt3a/

*Figure 2 continued on next page*

*Figure 2 continued*

Dnmt3b DcKO littermates after 6 months of DMBA/TPA treatment. Right, time of appearance of tumors represented as percentages in wild-type and Dnmt3a/Dnmt3b DcKO animals, and number of skin tumors after 3 or 6 months of treatment with DMBA/TPA. (C–D) Number of tumors (left) and time of appearance (right) expressed as percentages, in wild type, Dnmt3a-cKO and DcKO animals after 6 months of DMBA/TPA treatment. (E) Histopathological analysis of the different subsets of skin tumors that appeared after DMBA/TPA treatment of wild type or DcKO animals. (F) Representative images of metastatic nodules identified only in a percentage (33%) of the lungs of DcKO animals, scale bar = 100 μm.

The following figure supplements are available for figure 2:

**Figure supplement 1.** Deletion of Dnmt3b does not affect epidermal and hair follicle homeostasis.

**Figure supplement 2.** Dnmt3b-KO and wild-type skin tumors are histologically indistinguishable.

**Figure supplement 3.** Dnmt3b-KO tumors do not show changes in proliferation or apoptosis compared to their wild-type counterparts.

**Figure supplement 4.** The combined deletion of Dnmt3a and Dnmt3b does not affect epidermal homeostasis.

**Figure supplement 5.** Squamous cell carcinomas in Dnmt3a/Dnmt3b double KO mice express lower levels of epithelial markers compared to wilt-type tumors.

**Figure supplement 6.** The combined deletion of Dnmt3a and Dnmt3b favors the development of skin tumors with features of spindle cell carcinomas.

upregulated in Dnmt3a-cKO tumors (*Figure 3D*, *Supplementary file 1*). Among these, the most upregulated ones encoded the key pro-adipogenic transcription factors PPAR-α and PPAR-γ, which promote adipocyte differentiation and the expression of genes involved in fatty acid metabolism, and which are not expressed in homeostatic epidermal cells (*Fajas et al., 2001*). The role of these transcription factors in cancer is still poorly understood, although they tend to be upregulated in many types of human tumors (*Fajas et al., 2001*). Importantly, PPAR-γ was upregulated at the RNA and protein levels in all the Dnmt3a-cKO sequenced tumors (*Figure 3E* and *Figure 3—figure supplement 4A–B*). Interestingly, the expression of PPAR-γ has been extensively reported to be under epigenetic control by repressive mechanisms such as H3K9 methylation and DNA methylation (*Wang et al., 2013*; *Zhao et al., 2013*).

To further dissect the early molecular changes that might result in the tumor-suppressing role of Dnmt3a in the epidermis, we did a short (6-week long) DMBA/TPA carcinogenesis treatment (*Figure 4A*). We then FACS-isolated Itga6^brightCD34^neg cells, consisting mostly of epidermal basal cells (IFE), and hair follicle stem cells (Bulge; Itga6^brightCD34^pos) from pre-cancerous back skin of wild-type or Dnmt3a-cKO animals for RNA-seq analysis (*Supplementary file 2*). After this short DMBA/TPA treatment, most of the upregulated genes in epidermal cells (IFE) were already predominantly linked to lipid metabolism and cell proliferation, whereas they related mostly to cell *proliferation*, and *Wnt signaling* in bulge stem cells (*Figure 4—figure supplement 1A–B*). Interestingly, we did not observe a diminished expression of genes regulating apoptosis, as we did in tumor cells. Hence, these results suggest that most of the transcriptome changes observed in tumors upon deletion of Dnmt3a occur early, and that the transition from the pre-cancerous epithelium to tumor growth occurs subsequently by bypassing apoptosis.

## Dnmt3a binds to enhancers of epidermal differentiation genes that are DNA methylated and hydroxymethylated

Dnmt3a is responsible for establishing and maintaining the levels of both 5-mC and 5-hmC around enhancers and promoters (*Colquitt et al., 2014*; *Yang et al., 2016*). In addition, Dnmt3a directly methylates the center of its target enhancers resulting in their subsequent hydroxymethylation via Tet2 in human epidermal keratinocytes (*Rinaldi et al., 2016*). To study which targets are regulated directly by Dnmt3a during transformation of murine epidermis, we performed ChIP-Seq for Dnmt3a in DMBA/TPA-treated pre-cancerous back skin epidermises from wild-type or Dnmt3a-cKO animals (*Figure 4A*). We also compared the ChIP-seq data obtained with MeDIP-seq and hMeDIP-seq performed on FACS-sorted tumor cells. The profiles of MeDIP-seq and hMeDIP-seq around regulatory

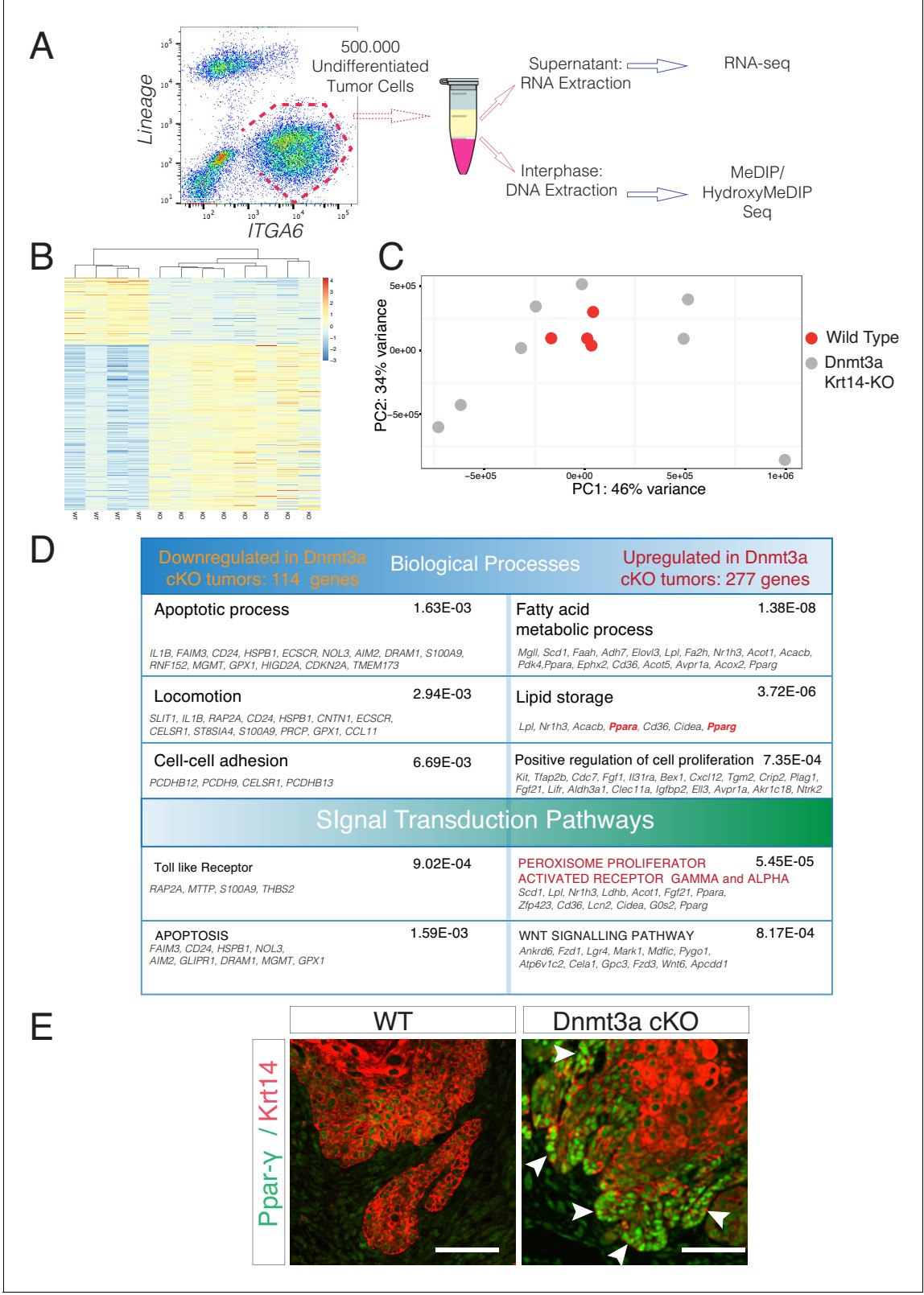

**Figure 3.** Deletion of Dnmt3a results in increased tumor heterogeneity, and upregulation of genes related to lipid metabolism. (**A**) Schematic representation of FACS sorting strategy to isolate both RNA and DNA from Itga6^pos cells within the tumors. (**B**) Heatmaps representing gene expression (rlog transformed values) of the 391 differentially expressed genes between wild type and Dnmt3a-cKO sorted tumor cells. (**C**) Two-dimensional principal-component analysis (PCA) of RNA-seq samples from wild-type (n = 4) and Dnmt3a-cKO (n = 8) Itga6^bright sorted tumor cells. (**D**)

*Figure 3 continued on next page*

*Figure 3 continued*

Gene ontology analysis using Genomatix Online Software of the 114 downregulated and 277 upregulated genes in Dnmt3a-cKO tumors, divided by biological processes and over-represented signal transduction pathways. (**E**) Immunofluorescence staining for Krt14 and PPAR-γ of skin tumors from wildtype and Dnmt3a-cKO animals.

The following source data and figure supplements are available for figure 3:

**Figure supplement 1.** RNA samples submitted for sequencing were obtained from tumors scored predominantly as squamous cell carcinomas in wild-type and Dnmt3a-cKO mice.

**Figure supplement 2.** Loss of Dnmt3a results in a reduction of apoptosis in skin tumors.

**Figure supplement 2—Source Data 1.** Data related to *Figure 3—figure supplement 2A–B*.

**Figure supplement 3.** DMBA/TPA treatment induces an increase in cellular cell proliferation in Dnmt3a-cKO animals.

**Figure supplement 4.** Dnmt3a-KO tumors express high levels of PPAR-γ.

**Figure supplement 4—Source Data 1.** Data related to *Figure 3—figure supplement 4B*.

regions (transcription start sites (TSS) and enhancers) agreed with published data (*Figure 5—figure supplement 1A*), and the CG content in our MeDIP-seq/hMeDIP-seq was highly enriched as compared to the input, both of which are measures of good quality data (*Figure 5—figure supplement 1B*).

We detected 16,483 genomic locations bound by Dnmt3a in wild-type animals, but only 64 in Dnmt3a-cKO, confirming the specificity of the Dnmt3a antibody (*Figure 4B* and *Supplementary file 3*). Of the bound regions in the wild-type epidermis, more than 20% corresponded to intergenic regions (*Figure 4B*). ChIP-Seq for H3K27ac using the same samples allowed us to identify 3097 intergenic regions enriched for H3K27ac that corresponded to active enhancers, 10% of which were bound by Dnmt3a in wild-type cells (*Figure 4A–C*, *Supplementary file 3*). Interestingly, proximity-based analysis revealed that the active enhancers bound by Dnmt3a predominantly corresponded to genes essential for *keratinocyte differentiation* and *transcriptional regulation*, such as *Evpl* (encoding for Envoplakin), *Ppl* (encoding for Periplakin), *Fos, Myc, Cebpa,* and *Fosl2* (*Figure 4C–D*), similarly to what we have previously reported in human epidermal keratinocytes (*Rinaldi et al., 2016*).

The active enhancers bound by Dnmt3a contained higher levels of DNA methylation and hydroxymethylation than those not bound by it (*Figure 5A,C*). Importantly, loss of Dnmt3a significantly reduced their DNA methylation and hydroxymethylation (*Figure 5A,C*). Intriguingly, a significant reduction in DNA methylation also occurred in enhancers not bound by Dnmt3a, albeit to a statistically significantly lesser extent than those directly targeted by Dnmt3a in wild-type cells (*Figure 5A, B*). Upon deletion of Dnmt3a, DNA hydroxymethylation was also significantly reduced in its target enhancers, and to a lesser extent in non-Dnmt3a-bound enhancers (*Figure 5C*). However, the ratio of 5-hmC levels at enhancers bound by Dnmt3a between wild-type and Dnmt3a-cKO epidermal cells is significantly higher as compared to the ratio of 5-hmC levels between the enhancers that are not normally bound by Dnmt3a (*Figure 5D*). This indicates that the presence of Dnmt3a correlates with significantly higher 5-hmC levels, likely because Dnmt3a provides 5-mC as a substrate for generating 5-hmC, as we have previously shown in human keratinocytes (*Rinaldi et al., 2016*).

## Dnmt3a binds to promoters of genes involved in cell proliferation and lipid metabolism to drive their DNA methylation

In addition to active enhancers, a significant proportion (19%) of the enriched regions for Dnmt3a corresponded to promoters/TSSs (*Figure 4B* and *Supplement file 3*). To understand if Dnmt3a was methylating these promoters, we overlaid the Dnmt3a ChIP-seq with the MeDIP-seq data. Notably, the promoters bound by Dnmt3a showed a strong and statistically significant loss of DNA methylation around the corresponding TSS (*Figure 6A*). The levels of DNA methylation were not significantly changed at promoters not bound by Dnmt3a (*Figure 6B*). Of note, Dnmt3a-target TSSs were not

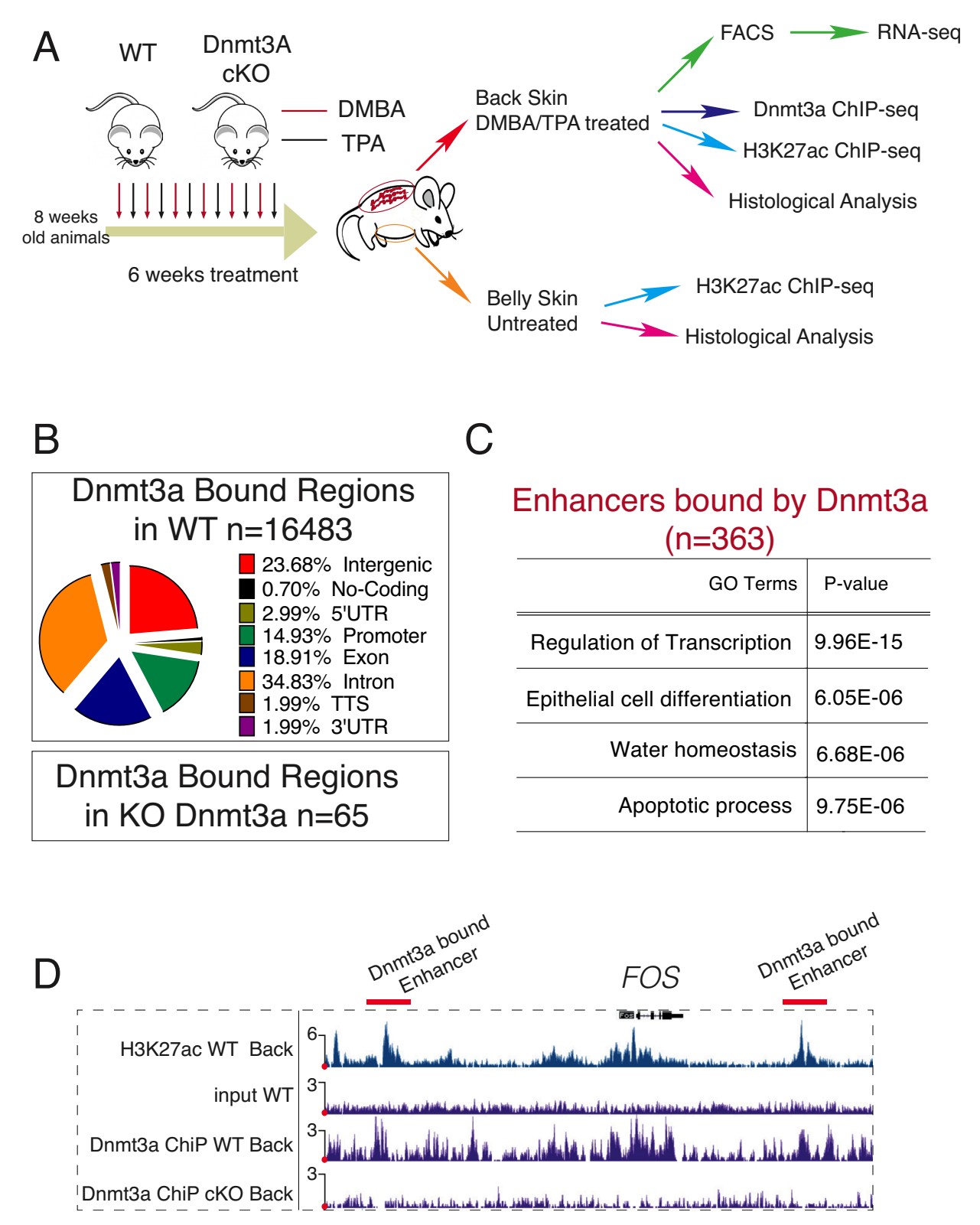

**Figure 4.** Dnmt3a binds a subset of enhancers in tumor cells. (**A**) Schematic representation of a short treatment of DMBA/TPA in wild-type and Dnmt3a-cKO animals. (**B**) Genomic localizations of Dnmt3a determined by ChIP-seq of Dnmt3a in epidermal cells isolated from wild-type animals after 6 weeks of DMBA/TPA treatment. (**C**) Gene ontology analysis of the 363 H3K27ac-enriched regions (located at least 4 kb away from the TSS) also

*Figure 4 continued on next page*

*Figure 4 continued*

bound by Dnmt3a in isolated epidermis from wild-type animals after 6 weeks of DMBA/TPA. (D) Screenshot of enhancers bound by Dnmt3a in DMBA/TPA-treated skin in the *FOS* locus. All tracks are normalized to the number of mapped reads.

The following figure supplement is available for figure 4:

**Figure supplement 1.** Deletion of Dnmt3a alters the expression of genes involved in proliferation, lipid metabolism, epidermal differentiation, and Wnt signaling, after 6 weeks of DMBA/TPA treatment.

enriched for 5-hMC (not shown). The loss of DNA methylation at the promoters/TSSs bound by Dnmt3a was also accompanied by a general increase in the transcription of these genes, measured by RNA-seq in the tumors (*Figure 6C*). Altogether, these data suggest that Dnmt3a directly represses the expression of a specific subset of genes by methylating their promoters/TSSs.

Interestingly, a GO analysis of the promoters bound by Dnmt3a indicated that they regulated the expression of genes predominantly involved in cell proliferation and lipid metabolism, consistent with our RNA-seq results (*Figure 6D* and *Supplement file 3*). Interestingly, Dnmt3a bound the promoters of *Ppar-α* and *Ppar-γ* in wild-type but not Dnmt3a-cKO epidermis (*Figure 6E* and *Supplement file 3*). Furthermore, 5-mC levels were lower in the TSS of the *PPAR-γ* gene in Dnmt3a-cKO as compared to wild-type tumors, indicating a DNA methylation-dependent mechanism of transcriptional repression (*Figure 6F*).

Consistent with a transcriptional derepression of the locus following loss of DNA methylation, PPAR-γ mRNA and protein levels were upregulated both in pre-cancerous interfollicular epidermis and in tumors lacking Dnmt3a, suggesting that the upregulation of PPAR-γ is acquired at the precancerous stage, even before overt tumors appear (*Figure 6G,H* and *Figure 6—source data 1*).

## Inhibition of PPAR-γ attenuates the protumorigenic effect elicited by the depletion of Dnmt3a

We next tested whether the increase in the expression of genes involved in lipid metabolism was required for the earlier onset of tumorigenesis and increased tumor burden in Dnmt3a-cKO mice. To this end, wild-type and Dnmt3a-cKO mice were subjected to the DMBA/TPA skin carcinogenesis protocol, but were separated into two cohorts, one treated topically with a PPAR-γ chemical inhibitor in combination with DMBA/TPA, and the other with the vehicle (*Figure 7A*-diagram) (*Sahu et al., 2012*; *Grabacka et al., 2006*). Interestingly, inhibition of PPAR-γ significantly delayed the onset of tumor appearance in Dnmt3a-cKO mice, and reduced the number of tumors developed by the Dnmt3a-cKO (*Figure 7B–C* and *Figure 7—source data 1*). However, the average size of the tumors was not affected by the inhibition of PPAR-γ (*Figure 7D*). Thus, inhibition of PPAR-γ could be a potential new therapy for cutaneous squamous cell carcinomas harboring low levels of Dnmt3a.

At last, using public available data from four different studies we determined that the expression of Dnmt3a is significantly reduced in squamous cell carcinomas and actinic keratosis, the premalignant stage of squamous tumors, compared to human healthy epidermis (*Figure 7—figure supplement 1* and *Figure 7—figure supplement 1—Source data 1*).

## Discussion

Dnmt3a modifies cytosine at CpG dinucleotides and is responsible for the proper differentiation of murine hematopoietic stem cells and murine neural stem cells (*Challen et al., 2011*; *Mayle et al., 2015*; *Shlush et al., 2014*). Recently, we and others have shown in human epidermal keratinocytes and murine olfactory sensory neurons, respectively, that Dnmt3a regulates gene expression by cooperating with Tet to maintain high levels of 5-hmC at enhancers (*Colquitt et al., 2014*; *Rinaldi et al., 2016*). Importantly, Dnmt3a is not only frequently mutated in human tumors (*Kim et al., 2013*) but is perhaps one of the first mutations to occur during tumorigenesis (*Shlush et al., 2014*). Using knockout mouse models, we now have demonstrated that Dnmt3a is also tumor-suppressive toward carcinogen-induced epidermal squamous neoplasia. Its loss not only accelerated the onset of tumors, but also increased tumor burden. However, once formed, Dnmt3a-deficient tumors grew, and progressed to carcinomas with the same kinetics and proportions, respectively, as their wild-

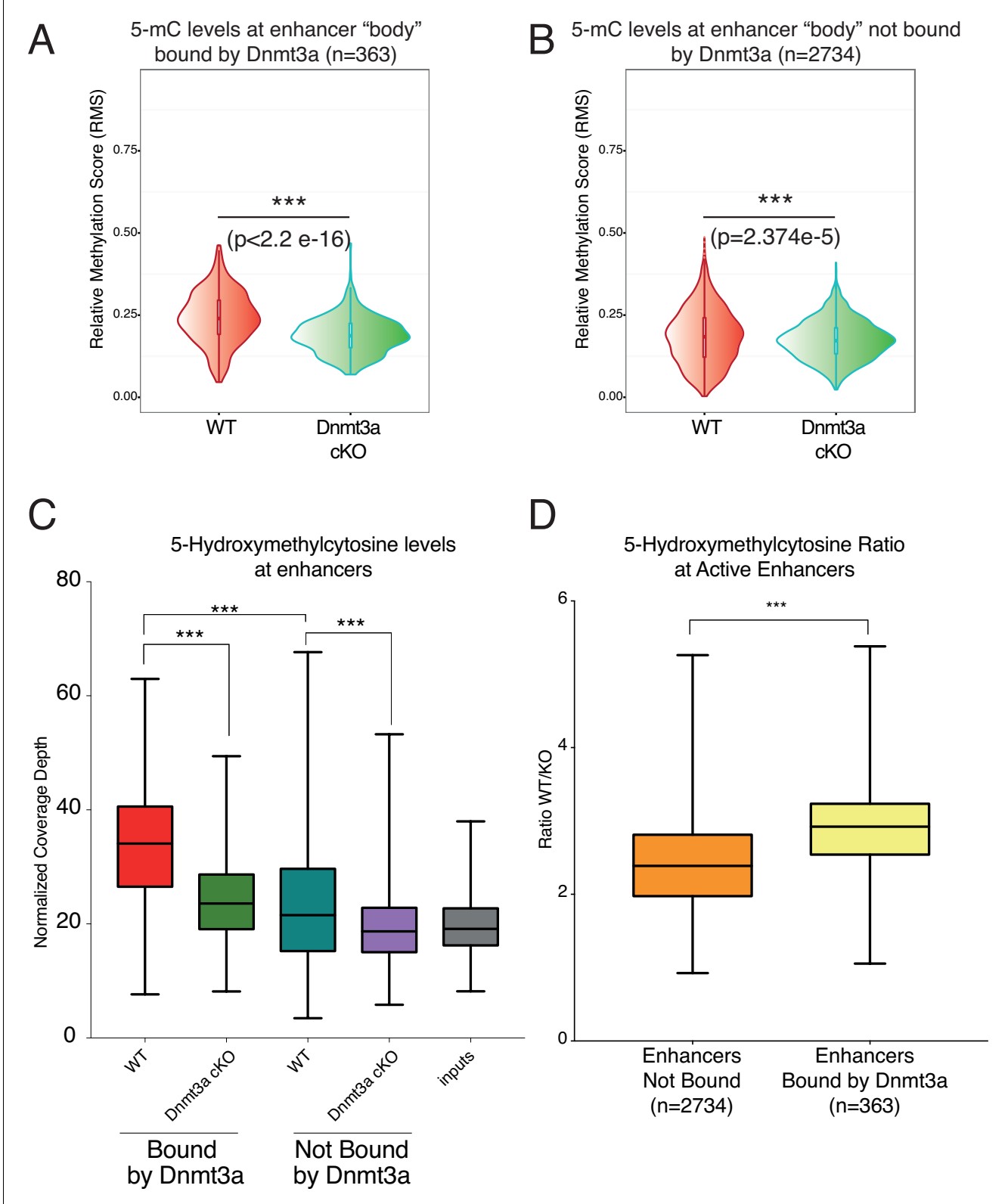

**Figure 5.** Depletion of Dnmt3a leads to loss of DNA methylation and hydroxymethylation around its target enhancers. (**A**) Relative methylation score (CpG count) measured around 363 enhancers bound by Dnmt3a (–5 kb, +5 kb) from independent biological replicates of FACS sorted tumor cells from wild type (n = 2) and Dnmt3a-cKO (n = 2) (p<2.2 × 10$^{-16}$). (**B**) Relative methylation score (CpG count) measured around 2734 enhancers not bound by Dnmt3a (–5 kb, +5 kb) from independent biological replicates of FACS-sorted tumor cells from wild-type (n = 2) and Dnmt3a-cKO (n = 2) animals
*Figure 5 continued on next page*

Figure 5 continued

(p=2.374e$^{-5}$). (C) Global levels of 5-hmC at enhancer center (–2Kb, + 2 Kb) were quantified using HOMER software in independent biological replicates of FACS sorted tumor cells from wild-type (n = 2) and Dnmt3a-cKO (n = 2) mice at enhancers bound or not by Dnmt3a. (D) Ratio between the 5-hmC levels at enhancers bound or not by Dnmt3a in wild-type and Dnmt3a-cKO tumor cells.

The following figure supplement is available for figure 5:

**Figure supplement 1.** MeDIP-seq and hMeDIP-seq analysis from sorted tumor cells.

type counterparts. Recent works have shown that the absence of Dnmt3a or Tet2 in hematopoietic stem cells predisposes to leukemia formation (*Yang et al., 2016*; *Rasmussen et al., 2015*), but that restoring the expression of Dnmt3a after the leukemia had developed did not revert the phenotype (*Yang et al., 2016*). That said, the role of Dnmt3a in tumorigenesis is tissue specific, since in the lung it does not affect tumor initiation but rather tumor progression (*Gao et al., 2011*). Interestingly, in the work of Gao et al., most of the changes in gene expression in Dnmt3a-depleted cells were attributed to alterations in gene body methylation, rather than at promoters. Conversely, in our model, we see significant changes at regulatory elements (i.e. promoters and enhancers) that lead to changes in gene expression in Dnmt3a-depleted epidermal tumors.

Our results, together with accumulating evidence from other groups, demonstrates a clear relationship between the levels of Dnmt3a–Tet–5-hmC and tumorigenesis: the inactivation of this axis in adult stem cells predisposes them to tumor initiation. Interestingly, a global reduction of 5-hmC is a hallmark of several cancer types, including squamous cell carcinoma, and is often correlated with poor prognosis (*Zhang et al., 2016*; *Liao et al., 2016*; *Shi et al., 2016*; *Ficz and Gribben, 2014*; *Lian et al., 2012*). Our results indicate that Dnmt3a drives the DNA-methylation, and subsequent hydroxymethylation, of a subset of enhancers that regulate the expression of genes involved differentiation. Conversely, Dnmt3a binds to, and DNA methylates, the promoters of cell proliferation and lipid metabolism genes to repress their expression. Interestingly, however, deletion of Dnmt3a does not result in changes in the specification of the different keratinocyte lineages in the skin (epidermis, hair follicles, and sebaceous glands), nor their homeostasis in adulthood. What is more, even the combined deletion of Dnmt3a and Dnmt3b, albeit significantly reducing overall DNA methylation levels, did not result in any skin phenotype even in aged mice. On the other hand, we have recently shown that Dnmt3a and Dnmt3b are necessary for the self-renewal and differentiation of primary human keratinocytes (*Rinaldi et al., 2016*). This apparent contradiction in the phenotypes observed might be due to the fact that our work with human keratinocytes relied on culturing the cells, which was recently shown in murine skin keratinocytes to induce a wound healing damaged-like reversible state that affects their epigenome and transcriptome (*Adam et al., 2015*). Thus, in vivo deletion of Dnmt3a might not be sufficient to alter the homeostasis of undamaged skin, but renders the epidermis more susceptible to situations of damage. Accordingly, the epidermis of Dnmt3a-cKO mice responded to the treatment of DMBA/TPA in a much more pronounced manner. This effect might not be specific to Dnmt3a. For instance, Dnmt1 is the DNA methyltransferase most highly expressed in epidermal cells and is responsible for about 70% of DNA methylation levels (*Li et al., 1992*). Its depletion causes a strong loss of self-renewal of primary human basal keratinocytes (*Sen et al., 2010*). However, its loss in murine epidermis leads to a mild increase in proliferation, and to a partial alopecia, only in very aged mice (*Li et al., 2012*). Intriguingly, these results also suggest that Dnmt1 and Dnmt3a/3b exert different functions in epidermal homeostasis, although future work will be required to study these putative differences in depth.

Notwithstanding the differences between in vivo and ex vivo studies, our results show that the genomic localization of Dnmt3a is very similar between intact murine keratinocytes and cultured human keratinocytes. Besides its localization at active enhancers of genes involved in epidermal differentiation, Dnmt3a also bound to, and methylated, promoters of genes that regulate cell proliferation and lipid metabolism to repress their expression. Among these genes, were the master regulators of lipid metabolism and adipogenesis PPAR-$\alpha$ and PPAR-$\gamma$. Interestingly, a number of recent studies have highlighted the importance of a persistent lipid metabolism in promoting tumor transformation, and tumor metastasis in colorectal, liver, breast and oral squamous carcinomas, as well as for enhancing chemoresistance of leukemia stem cells (*Pascual et al., 2017*; *Ma et al., 2016*;

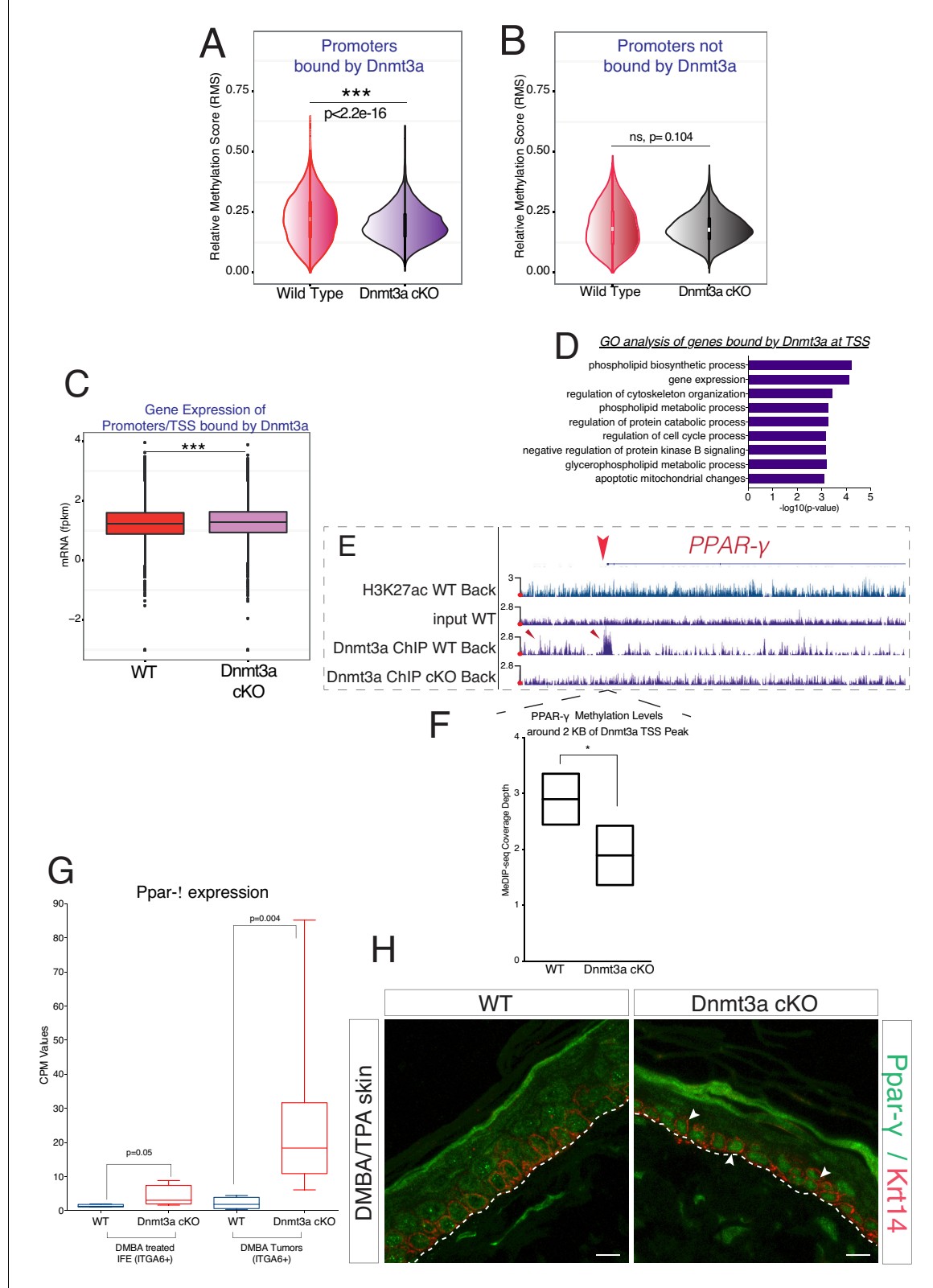

**Figure 6.** Dnmt3a binds and methylates a subset of promoters of genes involved in lipid metabolism in DMBA/TPA-treated epidermal cells. (**A**) Relative methylation score (CpG count) measured around active and silenced promoters bound by Dnmt3a (–5 kb, +5 kb) from independent biological replicates of FACS-sorted tumor cells from wild type (n = 2) and Dnmt3a-cKO (n = 2) animals. (**B**) Relative methylation score (CpG count) measured around promoters not bound by Dnmt3a (–5 kb, +5 kb) from independent biological replicates of FACS-sorted tumor cells from wild-type (n = 2) and

*Figure 6 continued on next page*

*Figure 6 continued*

Dnmt3a-cKO (n = 2) animals (p=0.104). (**C**) CPM (Counts per Million) values of genes bound at the TSS by Dnmt3a in DMBA skin tumors from wild-type or Dnmt3a-cKO animals. (**D**) Gene ontology analysis, using Enrichr online software, of the 3521 genes bound at their promoter by Dnmt3a. (**E**) Screenshot of *PPAR-γ* gene, with all tracks normalized. (**F**) Normalized methylation score measured around TSS of *Ppar-γ* (–1 kb to +1 kb) bound by Dnmt3a. (**G**) CPM (Counts per Million) values of PPAR-γ measured by RNA-seq in sorted Itga6$^{bright}$ cells from DMBA/TPA-treated IFE and from DMBA skin tumors in wild-type and Dnmt3a-cKO mice. (**H**) Immunofluorescence staining for Krt14 and PPAR-γ of DMBA/TPA-treated skin and skin tumors from wild-type and Dnmt3a-cKO animals.

The following source data is available for figure 6:

**Source data 1.** Data related to *Figure 6G*.

*Beyaz et al., 2016*; *Ye et al., 2016*). The upregulation of these transcription factors upon deregulation of Dnmt3a might predispose the epidermis to develop more tumors, suggesting that an intriguing mechanistic link between lipid metabolism and the epigenetic regulation of tissue homeostasis through DNA methylation, might exist. A recent large clinical association study has already pointed to this by establishing a correlation between the expression of obesity-related genes and changes in the content of DNA methylation (*Wahl et al., 2017*). Importantly, our results show that PPAR-γ is partially responsible for promoting tumorigenesis in Dnmt3a-deficient epidermis, which considering that human skin tumors express lower levels of Dnmt3a, might provide us with a new therapeutic antitumor avenue against squamous cell carcinomas.

## Materials and methods

### Chemical skin carcinogenesis

The work with mice was approved by the Ethical Committee for Animal Experimentation (CEEA) of the Scientific Park of Barcelona (PCB), and the Government of Catalunya. Inbred male or female *Dnmt3a* flox/flox (C57/Bl6) backcrossed to *Krt14-CRE*-YFP (C57/Bl6) for six to nine generations were used for all animal experiments. Chemically-induced skin carcinogenesis was performed as previously described (*Nassar et al., 2015*; *Abel et al., 2009*), with a slight modification to yield high-frequency SCCs in the C57/Bl6 genetic background. Briefly, the back skin of 8-week-old mice—at which time hair follicles are in their resting phase (telogen)—was shaved and treated with the mutagen 7,12-dimethylbenz[a]anthracene (DMBA; 200 μl of 0.25 mg/ml solution in acetone) and the pro-inflammatory and pro-proliferation agent 12-O-tetradecanoyl phorbol-13-acetate (TPA; 200 μl of 0.02 mg/ml solution in acetone) once weekly for 6 weeks. Specifically, DMBA was given on Monday and TPA always on the Friday of the same week. For short DMBA experiments, animals were sacrificed and back skins were processed 3 days after the sixth TPA application. For tumor formation studies, treatment continued twice weekly with TPA (200 μl of 20 μg/ml solution in acetone) for up to 20 weeks, or until the largest tumor of each mouse reached 1.5 mm diameter, at which point animals were sacrificed. In total, 12 wild type and 15 Dnmt3a-cKO tumors from 6 and 8 mice, respectively, were included for tumor analyses.

### PPAR-γ inhibitor treatment

The chemically-induced skin carcinogenesis was performed as previously described above (*Nassar et al., 2015*; *Abel et al., 2009*). We used the chemical 2-Chloro-5-nitro-N-phenylbenzamide (Sigma Aldrich: GW9662) described as potent PPAR- γ inhibitor. Briefly, the shaved dorsal epidermis of wild-type and Dnmt3a-cKO mice was treated twice a week topically with 200 μl of 100 nmoles of GW9662 solubilized in acetone. We applied the PPAR-γ inhibitor together with the first DMBA treatment, and subsequently administered it at every DMBA or TPA treatment. The PPAR-γ inhibitor was applied always 2 min before every administration of DMBA or TPA (*Sahu et al., 2012*; *Grabacka et al., 2006*).

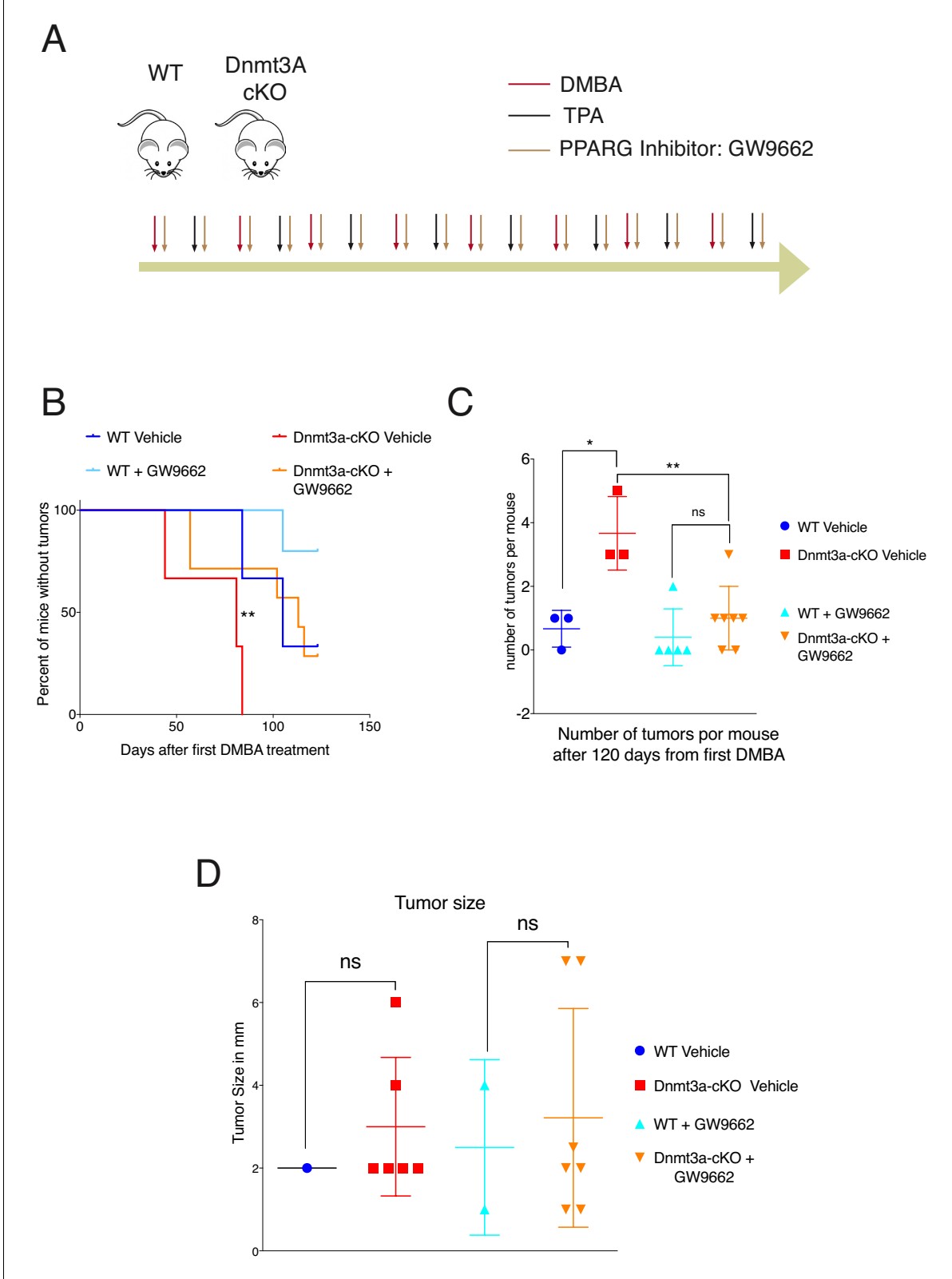

**Figure 7.** PPAR-γ inhibition revert the tumor initiation phenotype of the Dnmt3a-cKO. (**A**) Schematic representation of the DMBA/TPA orthotopic treatment together PPAR-γ inhibitor (Sigma GW9662) treatment onto wild-type and Dnmt3a-cKO animals. (**B**) Time of appearance, expressed in percentages of skin tumors on wild-type or Dnmt3a-cKO animals (vehicle and GW9662 treated): p=0.008, Chi-Square Test. (**C**) Number of skin tumors

*Figure 7 continued on next page*

*Figure 7 continued*

after 3 months of DMBA/TPA treatment plus GW9662 treatment, p=0.007 (Unpaired T-Test). (D) Tumors sizes expressed in millimeters (mm) after 3 months of DMBA/TPA plus GW9662 treatment.

The following source data and figure supplements are available for figure 7:

**Source data 1.** Data related to *Figure 7C*.

**Figure supplement 1.** The mRNA of Dnmt3a is downregulated in human cutaneous squamous cell carcinomas compared to normal human epidermis.

**Figure supplement 1—source data 1.** Data related to *Figure 7—figure supplement 1*.

## Single-cell preparation and FACS analysis

To isolate pre-cancerous epidermal cells following short DMBA/TPA treatment, back skins were dissected and processed to single-cell suspensions as previously described (*Jensen et al., 2010*). To purify tumor cells, DMBA/TPA-induced SCCs were mechanically dissociated using a McIlwain Tissue Chopper (The Mickle Laboratory Engineering Co. LTD). Minced tumor tissue was digested under agitation in serum-free EMEM medium without calcium containing 2.5 mg/ml Collagenase I (Sigma Aldrich, St. Louis, Missouri), and 0.75 mg/ml trypsin (Life Technologies) for 90 min at 37°C. Cells were pelleted, suspended in 1–2 ml of 0.25% pre-warmed trypsin/EDTA (Life Technologies) containing 100 µg/ml (Aldrich) per tumor, and incubated at 37°C for 2 min. Trypsin was inactivated by adding EMEM without calcium containing 10% chelated FBS. Cells were washed twice in PBS and filtered sequentially through 100 µm and 40 µm cell strainers.

For ChIP-seq, single-cell suspensions were cross-linked for 10 min at room temperature with 1% formaldehyde (methanol-free; Thermofisher, 28906) and quenched for 5 min to a final concentration of 0.125M of glycine. Cells were washed 2× with cold PBS and frozen at –80°C.

For flow cytometry analysis, epidermal or tumor cells were re-suspended at $1 \times 10^7$ cells/ml in PBS and labeled with CD49f-PE (clone NKI-GoH3, 1:200, AbD Serotec) and CD34-biotin (clone RAM34, 1:50, eBioscience) followed by streptavidin-APC (1:400, BD Biosciences). Tumor cell suspensions were additionally labeled with lineage-BV605 (CD31, clone 390; CD45, clone 30-F11; TER119, clone TER119; all 1:100) (Biolegend) to exclude stromal cell contamination. Both epidermal and tumor cells were positive for YFP due to the presence of the Rosa26-YFP allele in the mice.

Tumor cells (YFP$^{bright}$/lineage$^{neg}$ cells), pre-cancerous epidermis of interfollicular epidermis (YFP$^{pos}$/CD49f$^{high}$/CD34$^{neg}$ cells), and bulge hair follicle stem cells (YFP$^{bright}$/CD49f$^{high}$/CD34$^{pos}$ cells) were FACS-sorted using a BD FACSAria Fusion flow cytometer (BD Biosciences). Approximately, 3–$20 \times 10^4$ cells were sorted and lysed in 1 ml of TRIzol for RNA and DNA isolation. After adding 200 µl chloroform, samples were vortexed for 30 s and then centrifuged at 12,000 g to separate the RNA-containing supernatant from the organic phase. RNA was precipitated with 1× volume of isopropanol, washed twice with 70% ethanol, and then used for library preparation. The interphase of the TRIzol solution (after removal of the supernatant) was precipitated adding 1× volume of isopropanol, centrifuged for 1 hr at 4°C at 13,000 g, washed twice with ethanol, and digested overnight at 55°C with proteinase K (10 mg/ml) in TE 1× buffer. The following day, digested material was incubated 1 hr at 37°C with RNase A and purified using a conventional phenol/chloroform separation. The DNA pellet was quantified, and DNA was used for library preparation for MeDIP-seq and hMe-DIP-seq experiments.

## MeDIP and hMeDIP sequencing

Purified genomic DNA (250 ng) from tumor cells was sonicated to obtain fragments of 300–700 bp. Adaptors from the NEBNext Ultra DNA Library Prep Kit for Illumina were added to the fragmented DNA. DNA was denatured for 10 min at 99°C and cooled to avoid re-annealing. Fragmented DNA was incubated overnight with 1 µg of antibodies (5-methylcytosine, Abcam cat. # ab10805; 5-hydroxymethylcytosine, Active Motif, cat. # 39769, RRID: AB_10013602) previously cross-linked with 15 µl of Dynabeads Protein A (Life Technologies). Immunocomplexes were recovered using 8 µl for 2 hr. The following morning, DNA was washed three times for 10 min each, and purified DNA was

extracted using QIAquick MinElute (Qiagen). Amplified libraries were prepared using NEBNext Ultra DNA Library Prep Kit for Illumina (E7370L) following the manufacturer's instructions.

## RNA library preparation and sequencing

The libraries of total RNA from wild type and Dnmt3a-cKO tumors was prepared using the TruSeq-Stranded Total Sample Preparation kit (Illumina Inc.) according to the manufacturer's protocol. Each library was sequenced using TruSeq SBS Kit v3-HS, in paired end-mode with the read length $2 \times 76$ bp. A minimal of 137 million paired-end reads was generated for each sample run in one sequencing lane on HiSeq2000 (Illumina, Inc) following the manufacturer's protocol. Images analysis, base calling, and quality scoring of the run were processed using the manufacturer's software Real-Time Analysis (RTA 1.13.48) and followed by generation of FASTQ sequence files by CASAVA.

## RNA-seq data processing

RNA-seq datasets were pre-processed by removing both low-quality bases from the 3′- ends of the reads and adapter sequences using Trimmomatic (version 0.33) (*Bolger et al., 2014*). The trimmed reads were aligned to the mouse genome (UCSC mm10) using TopHat (version 2.0.13) (*Trapnell et al., 2009*), with default parameters and –g 5. Gene and transcript expression levels were quantified with HTSeq (version 0.6.1p1) (*Anders et al., 2015*). From the raw counts, counts per million (cpm) and fragments per kilobase of transcript per million mapped reads (fpkm) values were calculated. Differential expression analysis was performed using DESeq2 (*Love et al., 2014*) using a q-value cutoff of 0.05 and a fold-change cutoff of 1.5 to identify differentially expressed genes.

## Chromatin immunoprecipitation-sequencing (ChIP-seq)

ChIP was performed as previously described (*Morey et al., 2012*). Briefly, frozen pelleted were lysed in 1 ml ChIP buffer (150 mM NaCl, 10 mM Tris-HCl, 5 mM EDTA, 1% SDS, 0.5 mM DTT, and 1% Triton X-100) and sonicated for 30 min in a Bioruptor Pico (Diagenode). DNA fragments were de-cross-linked overnight at 65°C and checked with a bioanalyzer. After a DNA check, chromatin was diluted 1:5 with ChIP buffer with no SDS (150 mM NaCl, 10 mM Tris-HCl, 5 mM EDTA, 0.5 mM DTT, and 1% Triton X-100). Immunoprecipitation experiments for transcription factors used 30 μg of chromatin, and those for H3K27ac, 3 μg of chromatin. Antibodies (10 μg for Dnmt3a and 3 μg for H3K27ac) were incubated overnight with the chromatin in ChIP buffer. Immunocomplexes were recovered with 40 μl of protein A bead slurry (Healthcare, cat. # 17-5280-01). Immunoprecipitated material was washed three times with low-salt buffer (50 mM HEPES pH 7.5, 140 mM NaCl, 1% Triton) and 1× with high-salt buffer (50 mM HEPES pH 7.5, 500 mM NaCl, 1% Triton). DNA complexes were de-crosslinked at 65°C overnight, and DNA was then eluted in 50 μl of water using the PCR purification kit (QIAGEN). Antibodies used for ChIP were Dnmt3a (SantaCruz H-295; RRID: AB_2093990) and H3K27ac (Merck Millipore, cat. # 07–360; RRID: AB_310550). Libraries for sequencing were prepared using NEBNext Ultra DNA Library Prep Kit from Illumina (E7370L) following the manufacturer's instructions.

## ChIP-seq data processing

ChIP-seq datasets were aligned to the mouse genome build mm10 using BowTie (version 1.0.1) (*Langmead et al., 2009*); the parameters used were –k 1, –m 1, and –n 2. UCSC browser tracks (*Kent et al., 2002*) were created from the mapped bam file after converting it to bedGraph (normalized to 10 million reads) and subsequently bigWig format. Peak calling of Dnmt3A to determine regions of ChIP-seq enrichment over the background was done with the MACS version 1.4.1. Peaks of the methylation and hydroxymethylation datasets were determined similarly. For histone marks, MACS version two was used with parameters –broad, -q 0.01, and –g mm. ChIP-seq peaks were annotated using the annotatePeaks.pl script of the HOMER suite (version 4.6) (*Heinz et al., 2010*) using the UCSC mm10 annotation. The coverage depths of different ChIP-seq experiments at specified regions were also calculated using the annotatePeaks.pl script. This generated a normalized coverage value of different sequencing experiments at equally spaced bins spanning the region of interest. Bin size was set to 1 bp.

For the differential regulation analysis of MeDIP-seq data with replicates, common peaks were first determined among the replicates of the wild type and KO samples separately. A consensus

peakset was then created from the two common peaksets, and the read counts were calculated for all the peaks of the consensus peakset. DESeq2 (*Love et al., 2014*) was applied to calculate the differentially bound peaks using an adjusted p value of < 0.05.

## Immunofluorescence

Skin and tumors were isolated from mice, fixed in formalin 10% for 2 hr at room temperature, and embed in paraffin. Sections were cut and stained on glass coverslips. After deparaffinization sections were permeabilized with 0.5% Triton/PBS for 10 min, blocked with 10% goat serum, and stained overnight at 4°C with primary antibodies diluted in 1% goat serum. The morning after, sections were washed three times with PBS 1X with 10 min for each wash and stained with secondary antibody (1/1000). Nuclei were counterstained with DAPI (Sigma, D9542). Primary antibodies were anti-Dnmt3a (1:100, SantaCruz H-295:

RRID: AB_2093990), anti-PPAR-γ (1:100 Santa Cruz, sc-7196: RRID: AB_654710), and anti-keratin 14 (1:500, Biolegend SIG-3476: RRID:AB_10718041); secondary antibodies were anti-rabbit Alexa Fluor 488 (RRID: AB_141708) and anti-mouse Alexa Fluor 647 (1:500, Molecular Probes; RRID: AB_162542). For immunofluorescence staining anti-5-Methylcytosine (1:100, Abcam10805, clone 33D3: RRID: AB_442823), sections (after deparaffinization and before Triton incubation) were incubated for 15 min with 2N HCL to further denature DNA. Adipophilin (ADFP, ab37516, 1:100 dilution; RRID: AB_722641). Tunel Staining was performed using the Promega DeadEnd Tunel System following the manufacturer's instructions. Pictures were acquired using a Leica TCS SP5 confocal microscope.

## Statistical analysis

To compare tumor burden between genotypes, we used a T-Test with 95% confidence. To compare free tumor survival differences and anagen entry differences, we used a Chi-Square test. To compare Relative Methylation Score (RMS) levels and to compare normalized 5-hmC levels between wild type and Dnmt3a-cKO sorted tumor cells we used a paired Wilcoxon Test. The same paired-Wilcoxon test was used to measure differences in RNA expression.

## KI67 staining and quantification

Skin and tumor sections were stained after deparaffinization with KI67 (Abcam ab15580; RRID: AB_443209) for 60 min. After two washes, section were incubated with Power Vision Rabbit (Inmuno-Logic) for 45 min. Positive staining was revealed using a chromogen DAB for 5 min (Dako). Counterstain for hematoxylin was incubated for 3 min (Dako).

Stained sections were scanned using a high-resolution NanoZoomer 2.0 HT (Hamamatsu). KI67-positive nuclei in the interfollicular epidermis were measured using the TMarker software (*Schüffler et al., 2013*). Positive and negative nuclei for the staining were trained using the color deconvolution plugin and quantified using the cancer nucleus classification plugin. The total number of positive nuclei was normalized to the total number of nuclei in the area considered. Unpaired parametric T-test was used to measure statistical difference among groups and genotypes.

## Acknowledgements

The Spanish Ministry of Economy and Development (MINECO), and the Institute supported this project in the laboratory of SAB for Research in Biomedicine (IRB-Barcelona). IRB Barcelona is the recipient of a *Severo Ochoa Award of Excellence* from MINECO (Government of Spain). We are very grateful to the laboratories of Rudolph Jaenisch and Rafii Ahmed for providing us the Dnmt3a and Dnmt3b flox/flox animals. LR was sponsored by *La Caixa International PhD fellowship*. We thank all the core facilities at the IRB-Barcelona for their assistance in our work, and Veronica Raker for editing the manuscript. The raw data for every dataset included in the manuscript can be found at GEO (GSE87412).

# Additional information

## Funding

| Funder | Grant reference number | Author |
|---|---|---|
| Ministerio de Economía y Competitividad | National Grant, BFU2013-47990-P | Lorenzo Rinaldi<br>Alexandra Avgustinova<br>Debayan Datta<br>Guiomar Solanas<br>Salvador Aznar Benitah |
| European Research Council | STEMCLOCK (309502) | Lorenzo Rinaldi<br>Alexandra Avgustinova<br>Guiomar Solanas<br>Salvador Aznar Benitah |
| Fundación Botín | | Lorenzo Rinaldi<br>Alexandra Avgustinova<br>Debayan Datta<br>Guiomar Solanas<br>Salvador Aznar Benitah |

The funders had no role in study design, data collection and interpretation, or the decision to submit the work for publication.

## Author contributions

LR, Conceptualization, Formal analysis, Investigation, Methodology; AA, Investigation, Methodology; MM, Investigation; DD, Data curation, Formal analysis, Methodology; GS, Formal analysis, Supervision; NP, Formal analysis, Methodology; SAB, Conceptualization, Supervision, Funding acquisition, Investigation, Writing—original draft, Project administration, Writing—review and editing

## Author ORCIDs

Salvador Aznar Benitah, http://orcid.org/0000-0002-9059-5049

## Ethics

Animal experimentation: This study was performed in strict accordance with the recommendations in the Guide for the Care and Use of Laboratory Animals of the European Union. All of the animals were handled according to approved institutional animal care and use committee (CEEA) protocols (SAB-13-1522) of the Scientific Parc of Barcelona (PCB). The protocol was approved by the Committee on the Ethics of Animal Experiments of the Government of Catalunya.

# Additional files

## Supplementary files

• Supplementary file 1. This file contains information related to *Figure 3*. It shows the fold change difference of all RefSeq genes between *Dnmt3a*-cKO and wild type tumors in integrin alpha6-purified tumor cells. Please note that the negative values represent the genes that are expressed at higher levels in the *Dnmt3a*-cKO tumor cells.

• Supplementary file 2. This file contains information related to *Figure 4* and *Figure 4—figure supplement 1*. It shows the FPKM (Fragments Per Kilobase of transcript per Million mapped reads) values of all RefSeq genes obtained by RNA-sequencing in bulge cells (integrin alpha6bright/CD34+, BULGE) and interfollicular epidermis basal cells (integrin alpha6bright/CD34-, IFE) purified cells from *Dnmt3a*-cKO and wild type epidermis after 6 weeks of DMBA/TPA treatment.

• Supplementary file 3. This file contains information related to *Figure 4*, *Figure 5* and *Figure 6*. Spreadsheet1 contains the genomic coordinates and the related annotations of the Chip-Seq peaks of Dnmt3a in precancerous epidermis (after 6 weeks of DMBA/TPA treatment). Spreadsheet2 contains the list of genes bound at their promoter by Dnmt3a in precancerous

epidermis. Spreadsheet3 contains the genomic coordinates of the active enhancers not bound by Dnmt3a in precancerous epidermis. Spreadsheet4 contains the genomic coordinates of the promoters not bound by Dnmt3a in precancerous epidermis. Spreadsheet5 contains the genomic coordinates of the active enhancers bound by Dnmt3a in precancerous epidermis.

### Major datasets

The following dataset was generated:

| Author(s) | Year | Dataset title | Dataset URL | Database, license, and accessibility information |
|---|---|---|---|---|
| Rinaldi L, Benitah SA | 2017 | Dnmt3a associates with promoters and enhancers to protect epidermal stem cells from cancer | https://www.ncbi.nlm.nih.gov/geo/query/acc.cgi?acc=GSE87412 | Publicly available at the NCBI Gene Expression Omnibus (accession no: GSE87412) |

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
