## [Decision Letter]

Thank you for submitting your article "Dnmt3a associates with promoters and enhancers to protect epidermal stem cells from cancer" for consideration by *eLife*. Your article has been reviewed by three peer reviewers, one of whom is a member of our Board of Reviewing Editors, and the evaluation has been overseen by Fiona Watt as the Senior Editor. The reviewers have opted to remain anonymous.

The reviewers have discussed the reviews with one another and the Reviewing Editor has drafted this decision to help you prepare a revised submission.

Summary:

The reviewers agreed that understanding the role of Dnmt3a/b in skin carcinogenesis is an interesting and unresolved question. In particular, the authors convincingly and thoroughly show that loss of Dnmt3a in keratinocytes promotes increased tumorigenesis after DMBA treatment, which is intriguing. Transcriptional profiling and ChIP-seq data reveal genes altered by Dnmt3a.

Essential revisions:

1) The reviewers all raised concerns regarding the functional link between Dnmt and gene regulation in tumorigenesis. While the authors link the function to the suppression of PPAR and a lipid-type program, there are no functional data in the manuscript in support of this claim and, as mentioned below, this might be linked to a sebaceous tumor phenotype. Functional analysis of Dnmt targets is needed to extend this manuscript to more fully understand how Dnmt impacts tumor progression.

2) In Figure 1, they authors show that hairs regrow faster after DMBA treatment in Dnmt3a cKO mice. The authors suggested that "Dnmt3a might act as a break to maintain hair follicles in a dormant state". This potentially interesting phenotype has not been studied in detail and this information as it is might be a bit distracting.

3) It is clear that DNA methylation can affect non-transcriptional mechanisms such as genome stability and DNA repair mechanisms, and there are no studies presented to assess non-transcriptional effects of DNA methylation on tumorigenesis. Have the authors looked at mutation frequency and/or genomic stability in these tumors?

4) The authors argue that Dnmt3a selectively affects tumor initiation rather than tumor progression. These claims are questionable. Tumor initiation, which is difficult to study, refers to the first oncogenic conversion of a normal cell. In this study, the authors make these claims based on the macroscopic detection of tumors which could entirely depend on an effect on tumor progression. DMBA causes h-ras mutations and such tumors are highly sensitive to the presence of h-ras, indicating that this is the key initiation mechanism in the DMBA/TPA model. Did the authors observe an increase in h-ras mutations after DMBA application? Did the authors observe a difference in the repair of such mutations? Are skin tumors caused by activated k-ras also affected by Dnmt3a or is the effect specific to DMBA-induced tumors? In any case, to claim an effect on tumor initiation in epidermal stem cells, the authors would need to study earlier steps in tumorigenesis.

5) In the studies (Figure 3) where the authors isolated alpha6-high cells from WT and Dnmt3a-/- tumors for gene expression studies, it would be important to include images of the histology of each tumor as supplemental information. How did they ensure that all tumors were of the same type? Also, since the Dnmt3a tumors progress much faster, how can the authors rule out that the differences relate to different stages of tumorigenesis rather than different genotypes?

6) The findings of increased PPAR-γ and linked genes suggest the possibility that the authors may be detecting differentiation along the sebaceous lineage in these tumors. This is actually supported by the data indicating that a higher proportion of such tumors are found in the Dnmt3a ko mice. Again, having overview images of each of the tumors with staining for sebaceous markers would be helpful to rule this possibility out.

7) The characterization of apoptosis and link to Dnmt3a should be improved. Does Dnmt3a act after DNA damage and increase the number of cells undergoing apoptosis following DMBA? The authors should check for active caspase 3 after DMBA and show more convincing Tunel staining. The Tunel staining data (now as Figure 3—figure supplement 1) should be quantified with statistics, and also it would be important to make sure that the same types of tumors are compared across genotypes.

8) The initiation of hair wave domains after hair removal in second anagen is highly variable and 6 mice (as in Figure 1) is probably not sufficient to conclude that there is a difference between genotypes. Perhaps smaller domains were initiated and missed. When did the WT mice exhibit anagen patches? Plotting the time of first anagen patch might be more convincing (also this is not a major point in the paper).

9) The differences in gene expression reported in Figure 6 may be significant because of the high number of genes included in the comparison, but these differences are very small and not likely to be biologically important. Also, the differences in DNA methylation reported in Figure 6 are very small.

10) In the title and throughout the manuscript the authors refer to the study of epidermal stem cells. Cultured human keratinocytes under low calcium conditions cannot be accurately referred to as "stem cells". Furthermore, the stem cells within epidermal tumors would need to be studied in more detail to substantiate the claim that epidermal stem cells are particularly affected by Dnmt function. In Figure 4 alpha6+/CD34 is used as marker of interfollicular epithelial cells, which likely includes non-stem hair follicle keratinocytes.

11) The characterization of epidermal differentiation should be performed more carefully. Are the suprabasal epidermal layers normal in these mouse mutants? If deletion of Dnmt3a and Dnmt3b singly or in combination has little effect on epidermal differentiation during development, and epidermal homeostasis, indicating that these enzymes are dispensable for adult epidermal stem cell function, this finding would be an important one to emphasize in the manuscript. The findings in the current study are much more plausible and in line with previous studies suggesting that reversible DNA methylation does not seem to have a prominent role in gene regulation in normal tissues (it has clear roles in irreversible silencing such as imprinting, X-inactivation, transposon silencing, irreversible promoter silencing, but a role in re gene regulation has been difficult to demonstrate outside cell culture; see for example Bestor et al., PNAS 2015). There is very little data supporting the idea that methylation and demethylation are important regulators of cellular differentiation during embryonic development; the current manuscript is consistent with this view and contradicts the authors' recently published study. The authors need to discuss the divergence in their two studies more clearly and mention this key finding in the Abstract.

---

## [Author Response]

*Essential revisions:*

*1) The reviewers all raised concerns regarding the functional link between Dnmt and gene regulation in tumorigenesis. While the authors link the function to the suppression of PPAR and a lipid-type program, there are no functional data in the manuscript in support of this claim and, as mentioned below, this might be linked to a sebaceous tumor phenotype. Functional analysis of Dnmt targets is needed to extend this manuscript to more fully understand how Dnmt impacts tumor progression.*

We agree that providing functional analysis of Dnmt3a targets, in particular PPARg, would improve the mechanistic insight of our manuscript.

First, we would like to point out that we believe that the upregulation of lipid metabolism genes we observed is not a consequence of the higher incidence of sebaceous tumors in Dnmt3a-cKO mice. Although Dnmt3a-cKO mice develop more sebaceous adenomas than wild type mice, we made sure that this type of tumor was not used for any of the mechanistic studies included in our work. We apologize for not making this clear in the first version of our manuscript. To highlight this, we now have included a new figure (Figure 3—figure supplement 1) containing the histology of each tumor used of the RNA-sequencing studies, which was performed by our mouse pathologist to make sure that none was scored as a sebaceous adenoma. We have also performed immunostaining to detect the expression of ADFP to show that none of the tumors studied expressed this sebaceous marker (also included in Figure 3—figure supplement 1). In addition, we now include as Figure 3—figure supplement 4, the immunostaining to detect the expression of PPARg in each sample, showing that Dnmt3-cKO tumors (scored as squamous cell carcinomas, and not sebaceous adenomas) show high expression of PPARg. Furthermore, the RNA-seq data obtained from the short-term treatment with DMBA-TPA shows that basal IFE cells in Dnmt3a-cKO mice already start to express high levels of genes involved in lipid metabolism (including PPARg, as also shown in Figure 6) without any sebocyte differentiation.

Regarding the mechanistic studies: since we do not have PPARg conditional knockout mice available in our lab, we decided to use a well-characterized inhibitor of PPARg to address this issue. We repeated the DMBA-TPA chemical carcinogenesis protocol in wild type and Dnmt3a-cKO mice, and divided the mice into two cohorts, one treated with the PPARg inhibitor, and the other one with the vehicle. As shown in Figure 7, inhibition of PPARg significantly delayed the time of tumor appearance, and reduced the number of tumors, in Dnmt3a-cKO mice. Interestingly, the size of the tumors was not affected by inhibition of PPARg, suggesting that this TF is necessary for the early onset of tumorigenesis, but not for the growth of tumors once they are formed. These results therefore suggest that inhibition of PPARg could be considered as a new possible therapy for skin squamous cell carcinomas.

We would like to thank you for making this interesting suggestion, which we believe has significantly enhanced the quality of our work.

*2) In Figure 1, they authors show that hairs regrow faster after DMBA treatment in Dnmt3a cKO mice. The authors suggested that "Dnmt3a might act as a break to maintain hair follicles in a dormant state". This potentially interesting phenotype has not been studied in detail and this information as it is might be a bit distracting.*

We agree that this observation could distract the readers from the primary message of our work, which is to describe the role of Dnmt3a and Dnmt3b in skin tumorigenesis. We have therefore decided to remove panel A from Figure 1, and to study this interesting phenotype in more detail in the future.

*3) It is clear that DNA methylation can affect non-transcriptional mechanisms such as genome stability and DNA repair mechanisms, and there are no studies presented to assess non-transcriptional effects of DNA methylation on tumorigenesis. Have the authors looked at mutation frequency and/or genomic stability in these tumors?*

We believe this is a very interesting question. There are several publications suggesting that the epigenetic landscape of chromatin can have a major influence in the mutational load of tumors (for instance, Schuster-Böckler B and Lehner B. Chromatin organization is a major influence on regional mutation rates in human cancer cells. Nature 2012). However, these studies are correlative and do not provide any functional data. In this sense, before we reply directly to the issue raised above, we would like to point out that we have been developing for the last 4 years a large project designed to specifically determine whether the chromatin landscape (i.e. presence or absence of different chromatin modifications, and whether the chromatin is more accessible or closed) affects the mutational frequency and types of mutations in tumors. We have performed whole exome sequencing, ATAC-seq, and ChIP-seq of several chromatin marks in different mouse models in which we have deleted a specific epigenetic regulator, to precisely determine whether altering different chromatin marks has any effect over the regional mutation rates, types of mutations, and number and type of driver mutations, in DMBA/TPA-induced cutaneous carcinoma. This ongoing project has resulted in a vast amount of results, with really interesting but also quite complex conclusions. We therefore believe this analysis is beyond the scope of this current study, as it constitutes a story by itself. Because of the complexity of the analyses and results, adding just some of the data to this manuscript would not be sufficient to explain if and how epigenetic modifications, including DNA methylation, shape the mutational landscape of tumors.

However, we include as figures for the reviewers some results we have obtained regarding the potential role of genomic instability or mutational changes in the phenotype of Dnmt3a-cKO mice:

We have studied whether deletion of Dnmt3a results in increased genomic instability by looking at the DNA content in precancerous epidermis. To do so, we treated for six weeks three wild type and three Dnmt3a-cKO mice with DMBA/TPA. We then analyzed by FACS the DNA content in proliferative (Edu+) and non-proliferative (Edu-) Itga6brightCD34pos and Itga6brightCD34neg cells. Notably, we did not observe any significant change in the DNA content between Dnmt3a-cKO and wild type cells, suggesting that aneuploidy, or other signs of genomic instability, do not occur upon deletion of Dnmt3a in the epidermis (Figure 8 and Figure 9).

Author response image 1.**DOI:**
http://dx.doi.org/10.7554/eLife.21697.033

Author response image 2.**DOI:**
http://dx.doi.org/10.7554/eLife.21697.034

We also confirmed that the expression and localization of γ-H2AX, a marker of DNA double- strand breaks and general genomic damage, were the same in wild type and Dnmt3a-cKO tumors.

We took advantage of the high quality of our RNA-sequencing to start to understand if loss of Dnmt3a alters the mutational landscape in skin tumors. We obtained between 70-110 million reads per tumor; moreover, the reads were 125 nucleotides long, 2.5X more than the conventional 50 nucleotides; and also the sequencing was pair-ended. Importantly, we calculated that each nucleotide was covered in an average depth of 150-250x in each tumor (Figure 10).

Author response image 3.**DOI:**
http://dx.doi.org/10.7554/eLife.21697.035

These technical details gave us the possibility to analyze our samples using the GATK software (developed at the Broad Institute), which allows detecting variants, such as SNPs and other kinds of mutations using RNA-seq samples (Graw et al., 2015). As mentioned above, we are aware that whole exome/genome sequencing is the gold standard method to study mutations, but previous works have demonstrated that good quality and deep RNA- sequencing data can also be used with some confidence to identify exome variants (Piskol et al., 2013; Graw et al., 2015). Applying this method, we observed that Dnmt3a-cKO tumors showed a slight increase in the total number of mutations; however, this difference did not reach statistical significance (Figure 11)

Author response image 4.**DOI:**
http://dx.doi.org/10.7554/eLife.21697.036

We next tried to determine whether there were any differences in the types of mutations among the four bases, between wild type and Dnmt3-cKO tumors. As expected from previous works studying DMBA/TPA-induced tumors, the highest rate of mutations occurred on Thymine (Nassar et al., 2015). Interestingly, this analysis revealed that Dnmt3a-cKO tumors harbored a higher percentage of Cytosine to Thymine mutations compared to wild type tumors (Figure 12)

Author response image 5.**DOI:**
http://dx.doi.org/10.7554/eLife.21697.037

Although we do not know at the moment whether this change has any influence on the differences in the timing and number of tumors that Dnmt3a-cKO mice develop, we believe there might be a mechanistic explanation for it. Cytosine is one of the most-unstable bases, as it often spontaneously deaminates into Uracil (Gallinari et al., 1996). 5-methylcytosine also undergoes deamination, even more often than the native form, but instead it naturally mutates into Thymine instead of Uracil. In mammals Thymine-DNA glycosylase (TDG) is the enzyme primarily responsible for repairing the spontaneous deamination of 5-MethylCytosine, as part of the Base Excision Repair (BER) mechanism. TDG recognizes the 5mC>T mutation, excises the damaged base, and subsequently initiates the process to restore the native Cytosine (Jacobs et al., 2012). Interestingly, TDG interacts with many factors such as CBP/p300, ERα, but most importantly, with Dnmt3a (Li et al., 2006). In fact, the interaction with Dnmt3a positively regulates the glycosylase activity of TDG. Recent works have underscored the strong relationship between the DNA methylation machinery and the enzymatic activity of TDG (Cortellino et al., 2012; Dalton et al., 2013; Shen et al., 2013; Muller et al., 2014). Thus, altogether we believe that the depletion of Dnmt3a leads to an impairment of the activity of TDG glycosylase, resulting in the defective repair of 5mC>T mutations.

Although exome sequencing of Dnmt3a-cKO HSCs and leukemias has been published, the raw data is not available (or we have not been able to find it). Thus, unfortunately, we have not been able to compare our data with theirs to determine whether this difference in C-T mutations is generally associated to the loss of Dnmt3a (Mayle et al., 2015; Celik et al., 2015). In any case, and to conclude, we therefore believe that loss of Dnmt3a does affect the mutational landscape of DMBA/TPA-driven tumors. However, studying if and how these differences impact tumorigenesis is turning out to be very complex and is part of a large separate study in our lab (as mentioned above); hence, we believe that addressing this issue is beyond the scope of this current study.

4) The authors argue that Dnmt3a selectively affects tumor initiation rather than tumor progression. These claims are questionable. Tumor initiation, which is difficult to study, refers to the first oncogenic conversion of a normal cell. In this study, the authors make these claims based on the macroscopic detection of tumors which could entirely depend on an effect on tumor progression. DMBA causes h-ras mutations and such tumors are highly sensitive to the presence of h-ras, indicating that this is the key initiation mechanism in the DMBA/TPA model. Did the authors observe an increase in h-ras mutations after DMBA application? Did the authors observe a difference in the repair of such mutations?

These are very interesting questions and issues. Regarding the concept of tumor initiation versus tumor progression, we referred to differences in tumor initiation as the difference in the number of tumors that grow (or initiate) in Dnmt3a-cKO mice compared to wild type mice. On the other hand, we used the term progression to describe the transition from a benign lesion to a carcinoma. We believe these definitions are commonly accepted in the cancer field. According to these definitions, although Dnmt3a-cKO mice develop/initiate more tumors, they show the same proportion of benign and malignant lesions. Hence, loss of Dnmt3a does not favor this malignant conversion (progression) of squamous neoplasias.

However, as the reviewers point out, Dnmt3a-cKO mice might develop (or initiate) more tumors due to an increase in the percentage of cells mutated in Ras. In order to determine this, we would need to perform whole-exome sequencing of sorted epithelial tumor cells from a reasonably large cohort of wild type and KO tumors (for our other ongoing studies on the influence of chromatin on mutations we are sequencing at least 20 tumors per genotype). However, using the RNA-seq data included in the manuscript, and applying the GATK software described above, we could analyze mutations in HRAS and NRAS genes. As expected, we found mutations in HRAS and NRAS in wild type and Dnmt3a-cKO tumor cells. However, and being aware that the number of samples sequenced precludes reaching definitive conclusions, the percentage of mutations in H-RAS and N-RAS was not increased upon loss of Dnmt3a (Figure 13). In addition, the sites of mutations were the same between wild type and KO tumors in HRAS (T > A mutation), and NRAS (A >G mutations). Importantly, the HRAS mutations spotted by our analysis in Dnmt3a-cKO mice are precisely the same T>A mutation resulting in the activating Q61L transition, shown in two recent articles from the Blanpain and Balmain laboratories (Nassar et al., 2015 and McCreery et al., 2015).

Author response image 6.**DOI:**
http://dx.doi.org/10.7554/eLife.21697.038

In conclusion, our results suggest that Dnmt3a-cKO tumors do not contain more, or different, mutations in H-RAS and N-RAS. We therefore think it is unlikely that the shortened latency of tumor appearance, and increased tumor burden of Dnmt3a-cKO mice is due to differences in the mutations in these RAS genes

Are skin tumors caused by activated k-ras also affected by Dnmt3a or is the effect specific to DMBA-induced tumors?

In this manuscript we have only characterized the importance of Dnmt3a in carcinogen (i.e. DMBA)-induced tumors. We believe that addressing whether loss of Dnmt3a differently affects H-Ras- or K-Ras-driven tumorigenesis is very interesting, but beyond the scope of this study. In any case, previous work by the group of Rudolf Jaenisch showed that the deletion of Dnmt3a promotes lung tumor progression using a K-RAS mouse model, suggesting that Dnmt3a acts as a tumor suppressor regardless of which Ras gene is mutated

*In any case, to claim an effect on tumor initiation in epidermal stem cells, the authors would need to study earlier steps in tumorigenesis.*

As mentioned above, we think that our results are compatible with the conclusion that Dnmt3a-cKO mice initiate more tumors earlier than wild type mice. We believe this accurately describes the phenotype. Hence, we are not sure why this conclusion is not considered valid. We are however open to discussing any suggestion to perhaps more accurately describe our results.

Regarding tumor initiation, we believe we have in fact tried to provide some insights into the early steps of tumorigenesis with the short-term treatments of DMBA/TPA (showed in Figure 4 and Figure 4—figure supplement 1). We have shown that the loss of Dnmt3a increases cell proliferation and the expression of lipid metabolism genes early after treatment with DMBA/TPA (Figure 3—figure supplement 2). However, we also show that, once formed, Dnmt3a- cKO tumors do not show increased cell proliferation compared to wild type tumors (Figure 1). Interestingly, a very similar conclusion has been reached by another work recently published by the laboratory of Margaret Goodell (Yang et al., 2016), where the authors state: “The inability to slow the leukemia by re-expression of DNMT3A suggests that DNMT3A loss is important for leukemia initiation but less so for maintenance”.

5) In the studies (Figure 3) where the authors isolated alpha6-high cells from WT and Dnmt3a-/- tumors for gene expression studies, it would be important to include images of the histology of each tumor as supplemental information. How did they ensure that all tumors were of the same type? Also, since the Dnmt3a tumors progress much faster, how can the authors rule out that the differences relate to different stages of tumorigenesis rather than different genotypes?

We have now included the histology for each tumor analyzed for gene expression as the new Figure 3—figure supplement 3. It is important to note that our mouse pathologist (Neus Prats, who is an author in the manuscript) performed the histological characterization of each tumor in a blind manner. In addition, we have now included the measurements of the time difference between tumor appearance and tumor collection for each mouse, showing the Dnmt3a-cKO tumors, although appearing before, do not progress faster than wild type tumors (new Figure 3—figure supplement 4). The animals were sacrificed for humane reasons when the biggest tumor reached 1.5cm (as described into the methods section), but the average time between tumor appearance and this endpoint was the same between wild type and Dnmt3a-cKO tumors.

6) The findings of increased PPAR-γ and linked genes suggest the possibility that the authors may be detecting differentiation along the sebaceous lineage in these tumors. This is actually supported by the data indicating that a higher proportion of such tumors are found in the Dnmt3a ko mice. Again, having overview images of each of the tumors with staining for sebaceous markers would be helpful to rule this possibility out.

We agree this was an important issue that we should have clarified better in our first version of the manuscript. First, our mouse pathologist did not score any of the tumors used for RNA- seq (or any subsequent analysis) as a sebaceous adenoma. We have also now included in the new Figure 3—figure supplement 3, the results of the immunofluorescence staining for the sebaceous gland marker Adipophilin (ADFP), for each of the sequenced tumors. We did not observe any protein expression of ADFP in any of the 12 tumors sent for sequencing. Accordingly, ADFP was not differentially expressed at the mRNA level in Dnmt3a-cKO tumors compared to wild type lesions based on our RNA-seq analysis. As a positive control, we stained for ADFP in a sebaceous adenoma, which was not sent for sequencing. We would also like to note that sebaceous adenomas are generally small and quite different macroscopically from squamous cell carcinomas, so we could easily discriminate them when collecting the tumors for FACS sorting and subsequent analyses.

At last, we have also included in the new Figure 3—figure supplement 4 the expression analysis of PPAR-γ by immunofluorescence in all the 12 sequenced tumors that our pathologist scored as squamous cell carcinomas. We observed that PPAR-γ is consistently higher in all the 8 Dnmt3a KO tumors compared to the 4 wild type tumors, both at the RNA level (measured by RNA-seq) and also at the protein level. In addition, PPAR-γ is increased, upon Dnmt3a deletion, in the interfollicular epidermis after the short term DMBA/TPA treatment previous to any tumor formation, suggesting that the PPAR-γ is an early event not associated to a general sebocyte differentiation.

In conclusion, we believe we now provide new conclusive data indicating that deletion of Dnmt3a results in a stronger expression of lipid metabolism genes prior to tumor initiation, that persists in squamous cell carcinomas.

7) The characterization of apoptosis and link to Dnmt3a should be improved. Does Dnmt3a act after DNA damage and increase the number of cells undergoing apoptosis following DMBA? The authors should check for active caspase 3 after DMBA and show more convincing Tunel staining. The Tunel staining data (now as Figure 3—figure supplement 1) should be quantified with statistics, and also it would be important to make sure that the same types of tumors are compared across genotypes.

To properly quantify the changes in apoptosis in the Dnmt3a-cKO mice, we have stained and quantified, in all the sequenced tumors, for TUNEL and active Caspase-3. The results show a statistically significant decrease of apoptosis in Dnmt3a-cKO tumors compared to wild type lesions. These results are now included in the new Figure 3—figure supplement 2.

*8) The initiation of hair wave domains after hair removal in second anagen is highly variable and 6 mice (as in Figure 1) is probably not sufficient to conclude that there is a difference between genotypes. Perhaps smaller domains were initiated and missed. When did the WT mice exhibit anagen patches? Plotting the time of first anagen patch might be more convincing (also this is not a major point in the paper).*

We agree with the reviewers that is not a major point of our work, and that perhaps we need to study this phenotype in more detail and with more mice. Therefore, we have decided to remove these results from the manuscript.

9) The differences in gene expression reported in Figure 6 may be significant because of the high number of genes included in the comparison, but these differences are very small and not likely to be biologically important. Also, the differences in DNA methylation reported in Figure 6 are very small.

We respectfully disagree that the differences we show might not be biologically relevant. The changes in gene expression that occur upon deletion of Dnmt3a do in fact have a strong biological consequence regarding the timing and number of tumors that mice develop upon DMBA/TPA treatment. However, we would like to point out that the main reason we performed these analyses was to determine whether binding of Dnmt3a to promoters affected in any way the expression of the bound genes. In order to better show this, we have divided the Dnmt3a-target genes (as determined by our ChIP-seq data) based on binding strength; that is, we have separated them into the top 10%, 20%, 50% genes whose promoters are most strongly bound by Dnmt3a. Interestingly, the top 10-20% genes most strongly bound by Dnmt3a showed a stronger upregulation in their mRNA expression upon deletion of Dnmt3a compared to the top 50% of genes bound by Dnmt3a, or to all bound genes (Figure 14). We believe these results therefore indicate that binding of Dnmt3a to promoters, correlates mostly with a downregulation of the expression of those target genes. Upon deletion of Dnmt3a, the average expression of those genes increases.

Author response image 7.**DOI:**
http://dx.doi.org/10.7554/eLife.21697.039

We also believe that the new data we have added showing that inhibition of PPAR-γ partially prevents the positive effects of the deletion of Dnmt3a over tumorigenesis in fact indicates that the differences in gene expression that we have observed are biologically relevant.

10) In the title and throughout the manuscript the authors refer to the study of epidermal stem cells. Cultured human keratinocytes under low calcium conditions cannot be accurately referred to as "stem cells". Furthermore, the stem cells within epidermal tumors would need to be studied in more detail to substantiate the claim that epidermal stem cells are particularly affected by Dnmt function. In Figure 4 alpha6+/CD34 is used as marker of interfollicular epithelial cells, which likely includes non-stem hair follicle keratinocytes.

We initially described the phenotype as a stem cell based one because of previous works by the groups of Cedric Blanpain (Lgr5-Cre, *Sox2*-Cre, K14-Cre), Roel Nusse (Axin-Cre), Elaine Fuchs (Lgr5-Cre, K14-Cre) and Valentina Greco (Keratin15-Cre, Lgr5-Cre), that strongly suggest that squamous tumors arise from different stem cells within the keratinocyte compartment of the skin. However, it is also true that the definition of mouse IFE stem cells is still under debate within the stem cell field. In addition, extensive work by the group of Fiona Watt has shown that squamous tumors can also arise from involucrin positive IFE cells, which are not stem cells. We therefore agree that using our current mouse models we have no way of discriminating between these possibilities (i.e. stem cell, progenitor cell, or differentiated cell origin). Hence, we have changed the title to the following one that we think more accurately describes our findings: ¨Loss of Dnmt3a and Dnmt3b does not affect epidermal homeostasis but promotes squamous transformation through PPAR-γ¨

11) The characterization of epidermal differentiation should be performed more carefully. Are the suprabasal epidermal layers normal in these mouse mutants? If deletion of Dnmt3a and Dnmt3b singly or in combination has little effect on epidermal differentiation during development, and epidermal homeostasis, indicating that these enzymes are dispensable for adult epidermal stem cell function, this finding would be an important one to emphasize in the manuscript. The findings in the current study are much more plausible and in line with previous studies suggesting that reversible DNA methylation does not seem to have a prominent role in gene regulation in normal tissues (it has clear roles in irreversible silencing such as imprinting, X-inactivation, transposon silencing, irreversible promoter silencing, but a role in re gene regulation has been difficult to demonstrate outside cell culture; see for example Bestor et al., PNAS 2015). There is very little data supporting the idea that methylation and demethylation are important regulators of cellular differentiation during embryonic development; the current manuscript is consistent with this view and contradicts the authors' recently published study. The authors need to discuss the divergence in their two studies more clearly and mention this key finding in the Abstract.

We agree that this is an important point. We have now added the following in the Discussion section: “Our results indicate that Dnmt3a drives the DNA-methylation, and subsequent hydroxymethylation, of a subset of enhancers that regulate the expression of genes involved differentiation. […] Notwithstanding the differences between in vivo and ex vivo studies, our results show that the genomic localization of Dnmt3a is very similar between intact murine keratinocytes and cultured human keratinocytes.”

In order to verify this, we have performed a comparative clonogenic assay of cells isolated from the bulge and the IFE of wild type and Dnmt3a-cKO mice. As it can be observed in Figure 15, Dnmt3a-cKO keratinocytes (irrespective of whether they come from the bulge or the basal IFE) show a significantly impaired clonogenic potential, just as we had previously reported in Rinaldi et al., for Dnmt3a-depleted primary human keratinocytes. Hence we believe that deletion of Dnmt3a renders primary basal keratinocytes more sensitive to situations of stress.

Author response image 8.**DOI:**
http://dx.doi.org/10.7554/eLife.21697.040

However, we would like to point out that previous work from the laboratory of Margaret Goodell has shown that deletion of Dnmt3a does result in changes in gene expression in adult murine hematopoietic cells. It is true that these differences result in mild differentiation phenotypes (that are mostly apparent when HSCs are serially transplanted into irradiated mice). But it is equally true that these mice end up developing leukemia, which is a clear alteration of tissue homeostasis (a phenotype that is exacerbated upon the combined deletion of Dnmt3a and Dnmt3b). Hence, we think that the statement: “…that reversible DNA methylation does not seem to have a prominent role in gene regulation in normal tissues (it has clear roles in irreversible silencing such as imprinting, X-inactivation, transposon silencing, irreversible promoter silencing, but a role in re gene regulation has been difficult to demonstrate outside cell culture”; is perhaps too strong, and that more in depth studies are needed to determine the exact role of these proteins, and importantly of DNA methylation, in the differentiation of different tissues. For instance we do not know if Dnmt3a and Dnmt3b compensate each other upon conditional deletion in tissues during embryonic development. Would their inducible deletion during adulthood result in the same phenotype? Perhaps the recently identified Dnmt3c, although specific to germ cells, could also compensate for the combined loss of Dnmt3a and Dnmt3b in somatic tissues. In addition, our results suggest that Dnmt1 is capable by itself to maintain some degree of DNA methylation in the long term in the absence of Dnmt3a and Dnmt3b. In fact, our manuscript contains some evidence supporting this last point. With our K14-Cre line, we have deleted Dnmt3a and Dnmt3b in the embryonic epidermis approximately at E13.5. However, when we looked at DNA methylation in old mice (over 70 weeks of age) with the combined deletion of Dnmt3a/Dnmt3b, we could still see some residual methylation present (Figure 2—figure supplement 4). Hence, we would like to avoid concluding that DNA methylation/demethylation is completely dispensable for epidermal homeostasis, since with our KO models we have not completely eliminated DNA methylation from the epidermis.